# Towards Provable Emergence of In-Context Reinforcement Learning

**Jiuqi Wang**
Department of Computer Science
University of Virginia
Charlottesville, VA 22903
jiuqi@email.virginia.edu

**Rohan Chandra**
Department of Computer Science
University of Virginia
Charlottesville, VA 22903
rohanchandra@virginia.edu

**Shangtong Zhang**
Department of Computer Science
University of Virginia
Charlottesville, VA 22903
shangtong@virginia.edu

## Abstract

Typically, a modern reinforcement learning (RL) agent solves a task by updating its neural network parameters to adapt its policy to the task. Recently, it has been observed that some RL agents can solve a wide range of new out-of-distribution tasks without parameter updates after pretraining on some task distribution. When evaluated in a new task, instead of making parameter updates, the pretrained agent conditions its policy on additional input called the context, e.g., the agent's interaction history in the new task. The agent's performance increases as the information in the context increases, with the agent's parameters fixed. This phenomenon is typically called in-context RL (ICRL). The pretrained parameters of the agent network enable the remarkable ICRL phenomenon. However, many ICRL works perform the pretraining with standard RL algorithms. This raises the central question this paper aims to address: Why can the RL pretraining algorithm generate network parameters that enable ICRL? We hypothesize that the parameters capable of ICRL are minimizers of the pretraining loss. This work provides initial support for this hypothesis through a case study. In particular, we prove that when a Transformer is pretrained for policy evaluation, one of the global minimizers of the pretraining loss can enable in-context temporal difference learning.

## 1 Introduction

Reinforcement learning (RL, Sutton and Barto [2018]) is a powerful machine learning paradigm for solving sequential decision-making problems via trial and error. In deep RL [Mnih et al., 2015, Schulman et al., 2015, 2017], a classic workflow is to first train an agent with an RL algorithm in a task and then deploy it in the same task. Once the task changes, the skills acquired by the agent will be obsolete, and a new agent must typically be trained for the new task by repeating the expensive procedure above. In-context RL (ICRL, Moeini et al. [2025b]), first coined by Laskin et al. [2023], has the potential to address this limitation. In short, ICRL enables the agent to learn in the forward pass of their neural network by adding some additional data, called the context, to the network's input. This learning in the forward pass occurs without any parameter updates. Specifically, in ICRL, the RL agent is pretrained on a set of pretraining tasks. After pretraining, the RL agent network's parameters stay fixed, and the agent gets tested on some new test tasks, which can significantly

differ from the pretraining tasks. For example, in Laskin et al. [2023], the test task is a bandit problem having opposite optimal arms to the bandit problems in the pretraining tasks. Despite this discrepancy between pretraining and test tasks, one observes that the agent's performance in the test task increases as the context grows, with the agent's parameters frozen. We recall that the context serves as the additional input to the agent network, and a representative example is the agent's historical observations and actions in the new task until the current time step [Laskin et al., 2023]. This performance improvement cannot be due to the hypothesis that the fixed pretrained parameters memorize the pretraining tasks because the test task can drastically differ from the pretraining tasks. The only plausible explanation seems to be that the pretrained parameters enable some reinforcement learning process in the forward pass of the network to process the information in the context and then output a good action. This inference-time RL process is called ICRL. Notably, by tasks, we include both policy evaluation and control problems. The former predicts the value function, while the latter searches for a performant policy.

ICRL has a diverse array of pretraining algorithms, which can be divided into supervised pretraining and reinforcement pretraining. The key idea of supervised pretraining is to perform imitation learning to train the agent network to imitate the behaviour of some existing RL algorithms. It is thus not surprising that the pretrained agent network behaves like an RL algorithm in the forward pass. We defer more discussion about supervised pretraining to Section 2. What is surprising is the reinforcement pretraining, where the agent network is some sequence model (e.g., Transformer [Vaswani et al., 2017]) and the pretraining algorithm is merely (a variant of) standard deep RL algorithms [Duan et al., 2016, Wang et al., 2016, Kirsch et al., 2022, Lu et al., 2023, Bauer et al., 2023, Grigsby et al., 2024a,b, Park et al., 2025, Xu et al., 2024]. Essentially, in reinforcement pretraining, the network is trained to output good action (for control tasks) or to make good value predictions (for policy evaluation tasks). It, however, turns out that the forward pass of the pretrained network behaves like an RL algorithm. We now give a few examples of reinforcement pretraining. Duan et al. [2016] use recurrent neural networks (RNNs) as the agent network and use trust region policy optimization [Schulman et al., 2015] as the pretraining algorithm. The proposed $RL^2$ algorithm performs impressively in multi-armed bandits, tabular environments, and visually rich domains. A closely related and concurrent work is by Wang et al. [2016], where they train RRNs with A3C [Mnih et al., 2016]. Grigsby et al. [2024a] propose Amago that trains a variant of Transformer [Vaswani et al., 2017] with customized REDQ [Chen et al., 2021], which works competitively in challenging robot benchmarks. Wang et al. [2025] train Transformers with temporal difference learning (TD, Sutton [1988]) and find that the pretrained Transformers perform well on unseen policy evaluation tasks. The pretraining algorithm is just a standard RL algorithm in all these examples. It gives rise to two important questions about the learned parameters:

  (i) How can the learned parameters enable ICRL in the network's forward pass?

  (ii) Why do those parameters emerge after using the standard RL algorithm for pretraining?

Both questions are challenging since answering them requires white-boxing the learned neural network's internals and understanding the dynamics of the pretraining algorithm. To our knowledge, no prior work can answer both questions. In addition, given the diversity of the pretraining algorithms, having a single answer to all pretraining algorithms is unlikely. We present a case study following Wang et al. [2025] in this work.

Wang et al. [2025] answer Question (i) for policy evaluation tasks. In particular, they use TD to pretrain a Transformer for policy evaluation tasks on multiple randomly generated MDPs. After the pretraining converges, they study the converged parameters. Surprisingly, Wang et al. [2025] prove that with the converged parameters, the layer by layer forward pass of their Transformer is equivalent to iteration by iteration updates of TD. In other words, the converged Transformer performs policy evaluation in its forward pass by implementing TD in the forward pass. However, Wang et al. [2025] only partially answer Question (ii). In particular, they show thay the converged parameters belong to an invariant set of the the pretraining algorithm (i.e., TD). By an invariant set, we mean a set of parameters such that once the pretraining algorithm enters, it remains in the set forever (in expectation). For example, if the parameter is in $\mathbb{R}^d$, then a trivial invariant set is just $\mathbb{R}^d$ itself. The invariant set characterized by Wang et al. [2025] is much smaller than $\mathbb{R}^d$ (so it is nontrival and contains valuable information). But Wang et al. [2025] fall short in three aspects. First, their analysis considers only a single-layer Transformer. Second, their invariant set contains more parameters than the converged parameters. Third, whether the pretraining algorithm will always converge to the

invariant set remains unclear. This work builds upon Wang et al. [2025] and provides finer answers to Questions (i) and (ii) by making the following three contributions.

1. (Inference Time Convergence) We prove that when a Transformer is parameterized with the converged parameters observed by Wang et al. [2025], the value prediction error made by the Transformer diminishes as the depth of the Transformer grows to infinity.

2. (Global Minimizer of Pretraining Loss) We prove that the converged parameters observed by Wang et al. [2025] is a global minimizer of a few pretraining algorithms, including both TD and Monte Carlo.

3. We empirically verify our theoretical insights with controlled environments mirroring our setup in theory.

## 2 Related Works

One can categorize the pretraining of ICRL into supervised pretraining and reinforcement pretraining. The key idea of supervised pretraining is imitation learning, or algorithm distillation [Laskin et al., 2023]. In supervised pretraining, a dataset is first constructed by running a few existing known RL algorithms on the pretraining tasks and then collecting all the trajectories generated during the execution of those RL algorithms. Notably, a trajectory starts from the initialization of the RL algorithm and runs until the RL algorithm converges, thus spanning multiple episodes. As a result, the rewards near the beginning of the trajectory are lower, and those near the end are higher. The trajectory thus demonstrates the RL policy's learning progress. Then, a sequence model is used to fit the trajectories in the dataset. Namely, for a trajectory at time $t$, the input to the sequence model is all the data in the trajectory before time $t$. The target output of the sequence model is the action taken at time $t$. The sequence model is trained via an imitation learning loss to output the target action. The sequence model thus mimics the behaviour of RL algorithms. Notable ICRL works with supervised pretraining include Laskin et al. [2023], Liu and Abbeel [2023], Zisman et al. [2024], Shi et al. [2024], Huang et al. [2024a,b], Dai et al. [2024], Zisman et al. [2025], Polubarov et al. [2025]. Overall, we argue that in supervised pretraining, it is not surprising that the sequence model behaves like an RL algorithm in the forward pass because it was trained to do so. Lin et al. [2024] provide a theoretical analysis of supervised pretraining. In particular, they prove that with supervised pretraining, the pretrained sequence model behaves like the RL algorithms used to generate the dataset indeed. Lin et al. [2024] also provide some parameter construction of Transformers, such that the Transformer with those parameters can indeed implement some RL algorithms in the forward pass, including LinUCB [Chu et al., 2011], Thompson sampling [Russo et al., 2018], and UCB-VI [Azar et al., 2017]. However, the parameter construction in Lin et al. [2024] is overly complicated, and no evidence shows the constructed parameters are learnable. Our work is different from Lin et al. [2024] in that (1) we study reinforcement pretraining, whose dynamics are entirely different from supervised pretraining; (2) the Transformer parameters we study are learnable by practical and standard RL algorithms, which we empirically demonstrate.

In terms of reinforcement pretraining, the closest to our work is Park et al. [2025], which studies ICRL with large language models (LLMs) and proposes a regret-loss for reinforcement pretraining. They prove by construction that the global optimizer of their regret-loss implements the FTRL algorithm [Shalev-Shwartz and Singer, 2007]. While we classify Park et al. [2025] as reinforcement pretraining, it certainly deviates from standard RL algorithms (cf. the reinforcement pretraining algorithms in Section 1). Our work differs from Park et al. [2025] in that (1) the pretraining algorithm is different (so the pretraining dynamics are entirely different); (2) Park et al. [2025] only study single-layer Transformers, while our results apply to multi-layer Transformers; (3) No evidence exists to support that the parameters studied by Park et al. [2025] are learnable empirically, while the Transformer parameters we study are learnable by practical and standard RL algorithms, which we empirically demonstrate. Tarasov et al. [2025] build on top of algorithm distillation [Laskin et al., 2023] and propose to optimize the Transformer with RL objective. They report considerable improvement (30%) over the original algorithm distillation baseline on the testbeds. However, their work is entirely empirical and lacks theoretical insights. Moeini et al. [2025a] investigate ICRL with a safety constraint. They observe that reinforcement pretraining produces a better policy than the supervised pretraining counterpart in the safe ICRL domain.

ICRL falls into the category of black box meta RL. See Beck et al. [2025] for a more comprehensive treatment of meta RL. ICRL is also a special case of in-context learning (ICL, Brown et al. [2020]) if we use ICL to denote any learning paradigm that occurs in the inference time of the neural network. More commonly, ICL exclusively refers to inference-time supervised learning. A line of works studies the provable emergence of in-context supervised learning [Ahn et al., 2024, Zhang et al., 2024, Gatmiry et al., 2024]. The techniques we use in this paper are entirely different from those works because the dynamics of supervised learning and reinforcement learning are completely different.

## 3  Preliminaries

We use $I_n$ to denote an $n \times n$ identity matrix and $0_{m \times n}$ to denote an all-zero matrix in $\mathbb{R}^{m \times n}$ in this work. Given two vectors $x, y$, we will use $\langle x, y \rangle$ and $x^\top y$ to denote their inner product interchangeably.

### 3.1  Policy Evaluation

We consider an infinite-horizon Markov decision process (MDP, Puterman [2014]) with finite state space $\mathcal{S}$ and action space $\mathcal{A}$, a bounded reward function $r : \mathcal{S} \times \mathcal{A} \to \mathbb{R}$, a state transition probability function $p : \mathcal{S} \times \mathcal{S} \times \mathcal{A} \to [0, 1]$, an initial distribution $p_0 : \mathcal{S} \to [0, 1]$, and a discount factor $\gamma \in [0, 1)$. An agent implements a policy $\pi : \mathcal{A} \times \mathcal{S} \to [0, 1]$. Suppose the agent is at state $S_t$ at time step $t$, it outputs an action $A_t \sim \pi(\cdot \mid S_t)$, receives a reward $R_{t+1} \doteq r(S_t, A_t)$ and transitions to the next state $S_{t+1} \sim p(\cdot \mid S_t, A_t)$. When the policy $\pi$ is fixed, the MDP reduces to a Markov reward process (MRP), characterized by the tuple $\langle \mathcal{S}, p, p_0, r, \gamma \rangle$, where $p(s' \mid s) \doteq \sum_{a \in \mathcal{A}} \pi(a \mid s) p(s' \mid s, a)$ and $r(s) \doteq \sum_{a \in \mathcal{A}} \pi(a|s) r(s, a)$. The goal of policy evaluation is estimating the value function $v : \mathcal{S} \to \mathbb{R}$, defined as $v(s) \doteq \mathbb{E}\left[\sum_{t=0}^{\infty} \gamma^t R_{t+1} \big| S_0 = s\right]$. Since we will only consider policy evaluation tasks in this work, we will work with MRPs for simplicity. It is common to approximate $v$ with some parametric model $\hat{v}_w$ where $w \in \mathbb{R}^d$ is the weight. One of the simplest function approximators is arguably the linear model. Suppose there exists some feature mapping $\phi : \mathcal{S} \to \mathbb{R}^d$, a linear function approximation of $v$ is then $\hat{v}_w(s) \doteq \phi(s)^\top w \approx v(s)$. Note that $p$ defines a Markov chain. Assuming the chain is ergodic, there exists a unique and well-defined stationary distribution $\mu : \mathcal{S} \to [0, 1]$. Then, a common measure of the approximation error called the mean square value error (MSVE) can be well-defined as $\mathrm{MSVE}(v, v') \doteq \sum_{s \in \mathcal{S}} \mu(s)(v(s) - v'(s))^2$, for $v', v : \mathcal{S} \to \mathbb{R}$. Thus, one typically wishes to find $w^*$ such that $\mathrm{MSVE}(v, \hat{v}_{w^*})$ is minimized.

Monte Carlo (MC, Sutton and Barto [2018]) is a straightforward policy evaluation method. MC trains the function approximator towards the return, defined as $G(s) \doteq \sum_{t=0}^{\infty} \gamma^t R_{t+1} \mid S_0 = s$. We note that $\mathbb{E}[G(s)] = v(s)$ by definition. MC updates the weight vector as

$$w_{t+1} = w_t + \alpha_t (G(S_t) - \hat{v}_{w_t}(S_t)) \nabla \hat{v}_{w_t}(S_t),$$

where $\{\alpha_t\}$ is a sequence of learning rates. It is known that MC directly minimizes the MSVE [Sutton and Barto, 2018].

TD [Sutton, 1988, Sutton and Barto, 2018] is another fundamental policy evaluation algorithm. With linear function approximation, TD updates the weight vector iteratively as

$$w_{t+1} = w_t + \alpha_t (R_{t+1} + \gamma \hat{v}_{w_t}(S_{t+1}) - \hat{v}_{w_t}(S_t)) \nabla \hat{v}_{w_t}(S_t)$$
$$= w_t + \alpha_t (R_{t+1} + \gamma w_t^\top \phi(S_{t+1}) - w_t^\top \phi(S_t)) \phi(S_t),$$

where $\{\alpha_t\}$ is a sequence of learning rates. The term $(R_{t+1} + \gamma \hat{v}_{w_t}(S_{t+1}) - \hat{v}_{w_t}(S_t))$ is the TD error responsible for correcting the estimate. Unlike MC, TD forms the training target using its estimate, known as "bootstrapping" [Sutton and Barto, 2018].

The norm of the expected update (NEU) loss is an objective that gradient TD methods [Sutton et al., 2008, 2009] optimize. While it was originally defined exclusively for linear function approximation, we generalize it in our work to arbitrary function approximation. Given the weights $w$, we define NEU as $\mathrm{NEU}(w) \doteq \|\mathbb{E}[\delta \nabla \hat{v}_w(S)]\|$, where $\delta = G(S) - \hat{v}_w(S)$ for MC and $\delta = R + \gamma \hat{v}_w(S') - \hat{v}_w(S)$ for TD.

## 3.2 Linear Attention and Transformer

A canonical single-head self-attention [Vaswani et al., 2017] computes $\text{Attn}_{W_k, W_q, W_v}(Z) \doteq W_v Z \, \text{softmax}\left(Z^\top W_k^\top W_q Z\right)$, where $W_k, W_q, W_v$ denote the key, query, and value matrices, respectively, and $Z$ is the input prompt. The recent theoretical advancements of in-context reinforcement learning consider a linear attention variant, where the $\text{softmax}$ activation is replaced with an identity mapping [Park et al., 2025, Wang et al., 2025]. We consider the same linear self-attention structure and define $\text{LinAttn}(Z; P, Q) \doteq PZM\left(Z^\top QZ\right)$, where $Z \in \mathbb{R}^{(2d+1)\times(m+1)}$ is the input prompt, $P \in \mathbb{R}^{(2d+1)\times(2d+1)}$ is the reparameterization of $W_v$, $Q \in \mathbb{R}^{(2d+1)\times(2d+1)}$ is the reparameterization of $W_k^\top W_q$, and $M \in \mathbb{R}^{(m+1)\times(m+1)}$ is a fixed mask defined as $M \doteq \begin{bmatrix} I_m & 0_{m\times 1} \\ 0_{1\times m} & 0 \end{bmatrix}$. We delegate the first $m$ columns of $Z$ as the *context* and the $(m+1)$-th column for the *query*. The purpose of $M$ is to separate the query from the context during forward propagation. Having defined the linear attention, we can define the linear Transformer as a stack of linear self-attention layers. In an $L$-layer Transformer with parameters $\{P_l, Q_l\}_{l=0,\dots,L-1}$, the input prompt $Z_0$ evolves layer by layer as

$$Z_{l+1} \doteq Z_l + \text{LinAttn}_{P_l, Q_l}(Z_l) = Z_l + P_l Z_l M(Z_l^\top Q_l Z_l). \tag{1}$$

By the convention of Wang et al. [2025], we define $\text{TF}_L(Z_0; \{P_l, Q_l\}_{l=0,1,\dots L-1}) \doteq -Z_L^{(2d+1,m+1)}$ as the output of the $L$-layer Transformer, given input prompt $Z_0$. Notably, $Z_L^{(2d+1,m+1)}$ is the bottom-right element of the output embedding matrix, which is a scalar. The negative sign is in place to be consistent with previous work and simplify the notations in the proof.

## 3.3 In-Context Temporal Difference Learning

Wang et al. [2025] show that under certain parameterization of $P_l$ and $Q_l$, the layer-by-layer forward propagation of $\text{TF}_L$ is equivalent to an iteration-by-iteration execution of a batched version of TD algorithm on the context. In particular, let $S_0, R_1, S_1, \dots, R_m, S_m$ be a trajectory unrolled from a task. Suppose we wish to estimate $v(s_q)$ for some query state $s_q \in \mathcal{S}$. Then, using shorthands $\phi_i \doteq \phi(S_i)$ and $\phi_q \doteq \phi(s_q)$, we define for $l = 0, 1, \dots, L-1$

$$Z_0 \doteq \begin{bmatrix} \phi_0 & \dots & \phi_{m-1} & \phi_q \\ \gamma\phi_1 & \dots & \gamma\phi_m & 0_{d\times 1} \\ R_1 & \dots & R_m & 0 \end{bmatrix} \in \mathbb{R}^{(2d+1)\times(m+1)}, \tag{2}$$

$$P_l^{\text{TD}} \doteq \begin{bmatrix} 0_{2d\times 2d} & 0_{2d\times 1} \\ 0_{1\times 2d} & 1 \end{bmatrix}, Q_l^{\text{TD}} \doteq \begin{bmatrix} -I_d & I_d & 0_{d\times 1} \\ 0_{d\times d} & 0_{d\times d} & 0_{d\times 1} \\ 0_{1\times d} & 0_{1\times d} & 0 \end{bmatrix} \in \mathbb{R}^{(2d+1)\times(2d+1)}.$$

Let $\theta_L^{\text{TD}} \doteq \left\{P_l^{\text{TD}}, Q_l^{\text{TD}}\right\}_{l=0,\dots,L-1}$ represent the parameters compactly. Theorem 1 of Wang et al. [2025] proves that, given $Z_0$ defined in (2), it holds that $\text{TF}_L(Z_0; \theta_L^{\text{TD}}) = \langle\phi_q, w_l\rangle$, where $\{w_l\}$ is defined as $w_0 = 0$ and

$$w_{l+1} = w_l + \sum_{i=0}^{n-1} \left(R_{i+1} + \gamma w_l^\top \phi_{i+1} - w_l^\top \phi_i\right) \phi_i. \tag{3}$$

## 4 Inference Time Convergence

In this section, we establish the convergence and inference optimality of ICTD implemented by the $L$-layer linear Transformer as $L \to \infty$. Before we proceed to our analysis, we first need to redefine the input prompt $Z_0$ and the feature function $\phi$ to facilitate our proof in the rest of the paper. Let $n = |\mathcal{S}|$ and $s_0, s_1, \dots, s_{n-1}$ be the states in $\mathcal{S}$. We employ a one-hot feature function $\phi : \mathcal{S} \to \{0,1\}^n$ that maps each state $s$ to an $n$-dimensional vector where the $n$-th dimension is one and zero everywhere else. Using shorthands $\phi_i \doteq \phi(s_i)$ and $R_{i+1} \doteq r(s_i)$, we redefine $Z_0$ as

$$Z_0 \doteq \begin{bmatrix} \phi_0 & \dots & \phi_{n-1} & \phi_q \\ \gamma\sum_{j=0}^{n-1}\phi_j p(s_j \mid s_0) & \dots & \gamma\sum_{j=0}^{n-1}\phi_j p(s_j \mid s_{n-1}) & 0_{n\times 1} \\ R_1 & \dots & R_n & 0 \end{bmatrix} \in \mathbb{R}^{(2n+1)\times(n+1)}. \tag{4}$$

Now, the top $n$ rows of $Z_0$ are the *enumerations* of the states in $\mathcal{S}$, represented in one-hot encodings. Rows $n$ to $2n$ are now *expected* next features w.r.t $p$, discounted by $\gamma$. Hence, given $Z_0$ defined in (4), it holds that $\mathrm{TF}_L(Z_0; \theta_L^{\mathrm{TD}}) = \langle \phi_q, w_l \rangle$, where $\{w_l\}$ is defined as $w_0 = 0$ and

$$w_{l+1} = w_l + \sum_{i=0}^{n-1} \left( R_{i+1} + \gamma w_l^\top \sum_{j=0}^{n-1} \phi_j p(s_j \mid s_i) - w_l^\top \phi_i \right) \phi_i \tag{5}$$

by the virtue of (3). We employ $P \in [0,1]^{n \times n}$ to denote the transition probability matrix corresponding to $p$, where $P(i,j) \doteq p(s_j \mid s_i)$, and $\Phi \in \{0,1\}^{n \times n}$ to indicate the feature matrix, where the $i$-th row equals $\phi(s_i)$. It is obvious that $\Phi = I_n$. Likewise, we use $r \in \mathbb{R}^n$ to denote the vector representation of the reward function when it does not introduce ambiguity, where the $i$-th dimension is $r(s_i)$. Then, we have the following theorem.

**Theorem 1.** *Given a query state $s_q \in \mathcal{S}$ and constructing $Z_0$ as (4), it holds that* $\lim_{L \to \infty} TF_L(Z_0; \theta_L^{TD}) = v(s_q)$.

The proof is in Appendix B.1. Theorem 1 demonstrates that, under our formulation of $Z_0$ in (4), $\mathrm{TF}_L(Z_0, \theta_L^{\mathrm{TD}})$ converges to the *true value* $v(s_q)$ of the query state $s_q$ as the Transformer gets infinitely deep. This finding hints at the superiority of the parameterization $\theta_L^{\mathrm{TD}}$ from the policy evaluation point of view.

## 5 Global Minimizer of Pretraining Loss

Wang et al. [2025] discover that $\theta_L^{\mathrm{TD}}$ emerges after pretraining on some task and feature distributions using the proposed multi-task TD update. Their effort in explaining the emergence of $\theta_L^{\mathrm{TD}}$ includes proof that $\theta_L^{\mathrm{TD}}$ belongs to an invariant set of the multi-task TD update. However, their analysis is restricted to the single-layer case, yet the phenomenon is also observed in multi-layer linear Transformers. Previously, we justify that $\theta_L^{\mathrm{TD}}$ is desirable for in-context policy evaluation in Theorem 1. In this section, we investigate it from an optimization perspective and show that $\theta_L^{\mathrm{TD}}$ is one of the global minimizers of the pretraining loss.

Let $\Delta(r)$ denote the distribution of reward functions and $\Delta(p)$ denote the distribution of transition probabilities. Then, the tuple $\langle \Delta(r), \Delta(p) \rangle$ characterizes the MRP distribution. Let $\theta$ denote the (vectorization and concatenation of) parameters $\{(P_l, Q_l)\}_{l=0,\dots,L-1}$ of a Transformer of $L$ layers. We recall that in (4), the prompt $Z_0$ depends on the query feature $\phi_q$, as well as $p$ and $r$. To this end, we write $Z_0$ as $Z_0(p, r, \phi_q)$ to emphasize this dependency and use $Z_0(p, r, \phi(s))$ to denote the prompt by replacing the query $\phi_q$ with the feature $\phi(s)$ for some $s \in \mathcal{S}$. Wang et al. [2025] update $\theta$ iteratively as $\theta_{k+1} = \theta_k + \alpha_k \Delta^{\mathrm{TD}}(\theta_k)$, where

$$\Delta^{\mathrm{TD}}(\theta) \doteq \mathbb{E}_{s_q \sim \mu^p, s_q' \sim p(s_q), p \sim \Delta(p), r \sim \Delta(r)}[(r(s_q) + \gamma \mathrm{TF}_L(Z_0'; \theta) - \mathrm{TF}_L(Z_0; \theta)) \nabla_\theta \mathrm{TF}_L(Z_0; \theta)].$$

Here, $\mu^p$ denotes the stationary distribution of the transition dynamics $p$. We further define $Z_0' \doteq Z_0(p, r, \phi(s_q'))$ and $Z_0 \doteq Z_0(p, r, \phi(s_q))$. This motivates us to consider the Norm of Expected Update (NEU) loss, defined as $J(\theta) \doteq \|\Delta^{\mathrm{TD}}(\theta)\|_1$. Similar to Gatmiry et al. [2024], we enforce a sparsity assumption on the parameter space. Namely, we search $\theta$ over $\Theta$, defined as

$$\Theta \doteq \left\{ \left( P = \begin{bmatrix} 0_{2n \times 2n} & 0_{2n \times 1} \\ u & 1 \end{bmatrix}, Q = \begin{bmatrix} A & 0_{2n \times 1} \\ 0_{1 \times 2n} & 0 \end{bmatrix} \right) \middle| u \in \mathbb{R}^{1 \times 2n}, A \in \mathbb{R}^{2n \times 2n} \right\}^L.$$

Notably, in the parameter space $\Theta$, different Transformer layers can have different parameters (i.e., different $u$ and $A$). One special case of $\Theta$ is called the looped (autoregressive) Transformer, where all Transformer layers are forced to share the same parameters. We use $\Theta^{\mathrm{Looped}}$, defined as

$$\Theta^{\mathrm{Looped}} \doteq \left\{ \left( P = \begin{bmatrix} 0_{2n \times 2n} & 0_{2n \times 1} \\ u & 1 \end{bmatrix}, Q = \begin{bmatrix} A & 0_{2n \times 1} \\ 0_{1 \times 2n} & 0 \end{bmatrix} \right)^L \middle| u \in \mathbb{R}^{1 \times 2n}, A \in \mathbb{R}^{2n \times 2n} \right\},$$

to denote the parameter space (with sparsity constraints) of the looped Transformer we consider. It is easy to see that both $\theta_L^{\mathrm{TD}} \in \Theta$ and $\theta_L^{\mathrm{TD}} \in \Theta^{\mathrm{Looped}}$ hold. Although the evaluation of $\mathrm{TF}_L(Z_0; \theta)$ and $\mathrm{TF}_L(Z_0'; \theta)$ does not depend on the choice of the parameter space, the evaluation of $\nabla_\theta \mathrm{TF}_L(Z_0; \theta)$

does. When $\theta \in \Theta$, the gradient is taken w.r.t. $L$ different pairs of $(u, A)$. On the other hand, when $\theta \in \Theta^{\text{Looped}}$, the gradient is take w.r.t. a single pair of $(u, A)$.

The theorem below confirms that $\theta_L^{\text{TD}}$ is the global optimizer of the NEU loss for the looped Transformer when the number of layers grows to $\infty$.

**Theorem 2.** *Let the parameter space be $\Theta^{Looped}$. Then, it holds that $\lim_{L \to \infty} J(\theta_L^{TD}) = 0$.*

A few lemmas are in the sequl to prepare us to prove the above theorem.

**Lemma 1.** *For any query state $s_q$, transition probability $p$ and reward function $r$, it holds that*
$$\left| \mathbb{E}_{s_q' \sim p(s_q)} \left[ r(s_q) + \gamma TF_L(Z_0'; \theta_L^{TD}) - TF_L(Z_0; \theta_L^{TD}) \right] \right| \le \|r\|_\infty \gamma^L.$$

The proof is in Appendix B.2. Lemma 1 shows that the absolute value of the expected TD error decays at an exponential rate.

Define $X_l \doteq Z_l^{(1:2n,1:n)} \in \mathbb{R}^{2n \times n}$, the top left $2n$ by $n$ block of $Z_l$, and $Y_l \doteq Z_l^{(2n+1,1:n)} \in \mathbb{R}^{1 \times n}$, the bottom row of $Z_l$ except for the $(n+1)$-th entry to simplify notations. We furtherm define $x_l^q \doteq \begin{bmatrix} \phi_q \\ 0_{n \times 1} \end{bmatrix} \in \mathbb{R}^{2n}$ to represent the last column of $Z_l$ except for the last dimension. Likewise, we write $y_l^q$ to denote the bottom right entry of $Z_l$. Then, we can write $Z_l$ as $Z_l = \begin{bmatrix} X_l & x_l^q \\ Y_l & y_l^q \end{bmatrix}$.

**Lemma 2.** *Let $Z_l$ be the $l$-th embedding evolved by an $L$-layer linear Transformer parameterized by $\theta_L^{TD}$ following (1), where $Z_0$ is the input prompt defined in (4). Then, for $l = 0, 1, \ldots, L-1$, it holds that 1. $|y_l^q| \le (1 + \gamma^l)\|v\|_\infty$; 2. $\|Y_l\|_1 \le (\gamma + \gamma^l)\|v\|_\infty$ for all query state $s_q \in \mathcal{S}$, transition probability $p$ and reward function $r$.*

We leave the proof in Appendix B.3. Essentially, Lemma 2 implies that the norm of the evolving part of $Z_l$, i.e., the norm of $Y_l$ and $y_l^q$, is uniformly bounded because $\gamma^l \le 1$ for $l \ge 0$.

**Lemma 3.** $\left\| \nabla_\theta TF_L(Z_0; \theta_L^{TD}) \right\|_1 \le \frac{L^2 + L}{2} \nu + L\xi$ *for some constants $\nu$ and $\xi$ independent of $L$.*

The proof can be found in Appendix B.4. Lemma 3 shows the norm of the gradient term grows at a polynomial rate. We now have everything we need to prove Theorem 2 in Appendix B.5.

Recall that in multi-task TD, Wang et al. [2025] use the target $r(s_q) + \gamma TF_L(Z_0', \theta)$ to update the parameters of the Transformer, where $TF_L(Z_0', \theta)$ can be viewed as a (biased) proxy for $v(s_q')$. As $\mathbb{E}[r(s_q) + \gamma v(s_q')] = v(s_q)$, we conjecture that using the Monte Carlo return $G(s_q)$ as the target can also enable in-context policy evaluation capabilities in Transformers. In light of this, we propose the multi-task MC update that updates $\theta$ iteratively as $\theta_{k+1} = \theta_k + \alpha_k \Delta^{\text{MC}}(\theta_k)$, where

$$\Delta^{\text{MC}}(\theta) \doteq \mathbb{E}_{s_q \sim \mu^p, p \sim \Delta(p), r \sim \Delta(r)}[(G(s_q) - TF_L(Z_0; \theta))\nabla_\theta TF_L(Z_0; \theta)].$$

We again consider the NEU loss, defined as $J'(\theta) \doteq \left\| \Delta^{\text{MC}}(\theta) \right\|_1$. Remarkably, we also prove that $\theta_L^{\text{TD}}$ is the global optimizer of the NEU loss for our multi-task MC update as the Transformer's depth grows to infinity.

**Corollary 1.** *Let the parameter space be $\Theta^{Looped}$. Then it holds that $\lim_{L \to \infty} J'(\theta_L^{TD}) = 0$.*

We need the following lemma to assist us in proving the corollary.

**Lemma 4.** *For any query state $s_q$, transition probability $p$ and reward function $r$, it holds that*
$$\left| \mathbb{E}_{p,r} \left[ G(s_q) - TF_L(Z_0; \theta_L^{TD}) \right] \right| \le \|v\|_\infty \gamma^L.$$

The proof is in Appendix B.6. Lemma 4 is analogous to Lemma 1 in that it proves the norm of the expected approximation error of MC decays at an exponential rate. We can now prove Corollary 1 in Appendix B.7.

# 6 Experiments

We conduct empirical studies to verify our theoretical results. In particular, mirroring our theoretical setup, we investigate the following two questions:

1. Does the pretraining yield a Transformer that can perform in-context policy evaluation?
2. Do the converged parameters align with $\theta^{\mathrm{TD}}$?

We answer both questions for both multi-task TD and multi-task MC.

Following Wang et al. [2025], we employ Boyan's chain [Boyan, 1999] as our environment to construct the MRPs. Boyan's chain allows us to have full information and control over the environment, including analytically solving for the true values and the stationary distributions. Figure 3 shows an example of an $S$-state Boyan's chain. Adapting the technique of Wang et al. [2025], we randomly generate the reward and transition probability functions, preserving the topology of the chain to ensure its ergodicity. Our only simplifying modification is setting the stationary distribution as the initial distribution. The details of the task generation, including the distributions to sample $p$ and $r$, can be found in Algorithm 3.

Since our convergence results require $L \to \infty$, we construct a looped Transformer $\mathrm{TF}_L$ with $L = 30$. We find this depth yields consistent results without running into numerical instability. For each trial of the experiment, we generate 20,000 tasks for training with $\gamma = 0.9$. Within each task characterized by $p, r$, we generate a mini-batch of size $b = 64$. For multi-task TD, we unroll the MRP to generate a trajectory $S_0, R_1, S_1, R_2, \ldots, S_{b-1}, R_b, S_b$. Then, we update the parameter $\theta$ of the Transformer by

$$\theta_{k+1} = \theta_k + \frac{\alpha}{b} \sum_{i=0}^{b-1} \delta_i \nabla_\theta \mathrm{TF}_L(Z_0(p, r, \phi(S_i)); \theta_k),$$

where $\delta_i \doteq R_{i+1} + \gamma \mathrm{TF}_L(Z_0(p, r, \phi(S_{i+1})); \theta_k) - \mathrm{TF}_L(Z_0(p, r, \phi(S_i)); \theta_k)$. See Algorithm 1 for a more detailed description. Similarly, for multi-task MC, we unroll the MRP to generate a trajectory $S_0, R_1, S_1, R_2, \ldots, S_{b-2}, R_{b-1}, S_{b-1}$. For each $S_i$, we independently unroll the MRP for 200 steps to obtain a truncated return $\tilde{G}(S_i)$ beginning from that state. Admittedly, $\tilde{G}(S_i)$ introduces bias due to truncation. However, the bias is negligible as $\gamma^{200} \approx 7 \times 10^{-10}$. Multi-task MC updates the parameter by

$$\theta_{k+1} = \theta_k + \frac{\alpha}{b} \sum_{i=0}^{b-1} \left( \tilde{G}(S_i) - \mathrm{TF}_L(Z_0(p, r, \phi(S_i)); \theta_k) \right) \nabla_\theta \mathrm{TF}_L(Z_0(p, r, \phi(S_i)); \theta_k).$$

See Algorithm 2 for a more comprehensive illustration. We inherit most of the experimental settings from Wang et al. [2025]. See Table 1 for a comprehensive list of hyperparameters and training details.

## 6.1 In-Context Policy Evaluation Verification

Recall that one property of ICRL is that the agent's performance improves as the context gets longer without parameter updates. Based on this principle, we investigate the MSVE as a dependent variable of the context length for the converged Transformers. After each trial, we sample $k = 10$ novel tasks $T_1, \ldots, T_k$ from the task distribution as our validation set. Notably, Wang et al. [2025] theoretically and empirically proved that Transformers parameterized by $\theta^{\mathrm{TD}}$ can perform policy evaluation in context unrolled from an MRP. Thus, we employ the same context construction as in Wang et al. [2025] for *evaluation* instead of using the enumerated states and expected features that we used for *training* the Transformers. Suppose we need a context with length $m$ from $T_i$, we unroll $T_i$ for $m + 1$ steps and get $S_0^{(i)}, R_1^{(i)}, S_1^{(i)}, \ldots, S_{m-2}^{(i)}, R_{m-1}^{(i)}, S_{m-1}^{(i)}, R_m^{(i)}, S_m^{(i)}$. Then, we form the context as

$$C(i, m) \doteq \begin{bmatrix} \phi\left(S_0^{(i)}\right) & \phi\left(S_1^{(i)}\right) & \cdots & \phi\left(S_{m-1}^{(i)}\right) \\ \gamma\phi\left(S_1^{(i)}\right) & \gamma\phi\left(S_2^{(i)}\right) & \cdots & \gamma\phi\left(S_m^{(i)}\right) \\ R_1^{(i)} & R_2^{(i)} & \cdots & R_m^{(i)} \end{bmatrix} \in \mathbb{R}^{(2n+1) \times m}.$$

Given a query state $s_q$, we then define the input prompt $Z(i, m, s_q) \doteq [C(i, m) \quad x_q] \in \mathbb{R}^{(2n+1) \times (m+1)}$, where $x_q = \left[ \phi(s_q)^\top \quad 0_{1 \times (n+1)} \right]^\top$. Then, given a parameter $\theta$, we can evaluate the MSVE of the Transformer on $T_i$ with context length $m$ by $\mathrm{MSVE}(i, m) \doteq \sum_{s \in \mathcal{S}} \mu^{p^{(i)}}(s)(\mathrm{TF}_L(Z(i, m, s); \theta) - v(s))^2$, where $\mu^{p^{(i)}}$ means the stationary distribution with respect to the transition probabilities of $T_i$. We average the MSVEs across the validation tasks to get an

accurate estimation of the expected MSVE with context length $m$. We compute the averaged MSVEs with respect to an array of increasing $m$'s. We repeat our experiments for 20 trials for both multi-task TD and MC using the last checkpoint after pretraining. Figure 1 plots the mean and standard error of the averaged MSVE against the context length for multi-task TD and MC. The MSVE exhibits a clear decreasing trend as the context gets longer for both pretraining schemes. Therefore, we can confidently conclude that multi-task TD and MC can produce Transformers capable of in-context policy evaluation. The answer to question 1 is affirmative.

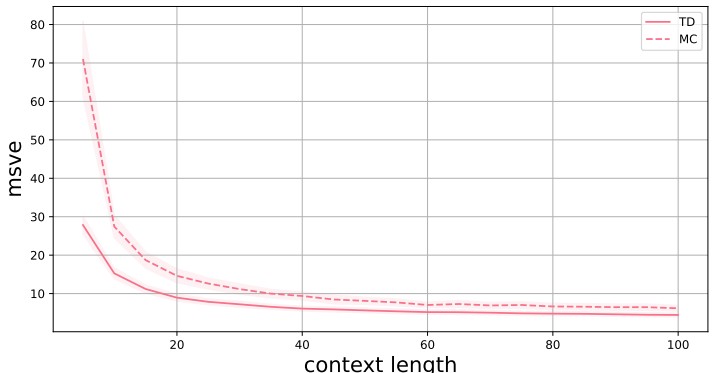

Figure 1: Mean and standard error of the averaged MSVEs against context lengths. The curves are averaged over 20 random trials. The shaded areas represent the standard errors.

## 6.2 Weight Convergence

In this section, we inspect the parameters of the Transformers after pretraining with multi-task TD and MC. Figure 2 displays the final $P$ and $Q$ matrices after pretraining. Despite mild noise, both methods produce the parameter configuration $\theta_L^{\text{TD}}$. Hence, the answer to question 2 is also affirmative. Surprisingly, even their noise patterns look very similar. Therefore, we speculate that the weights converge to the same place with multi-task TD and MC. Notably, the Transformer learns to implement in-context TD even after pretraining with multi-task MC. This observation testifies to the fact that the model does not merely imitate the reinforcement pretraining algorithm like in algorithm distillation for supervised pretraining. Instead, the model itself decides how to best learn from the context.

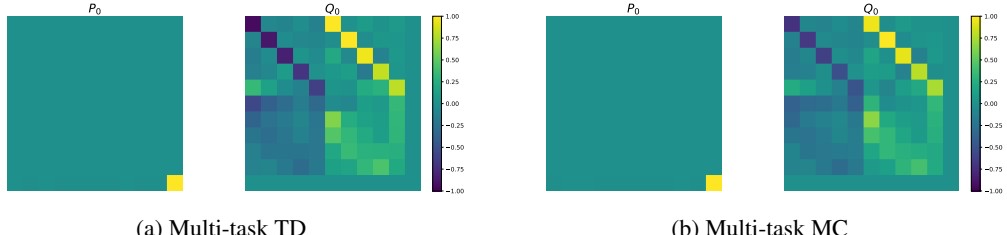

(a) Multi-task TD          (b) Multi-task MC

Figure 2: Mean Transformer parameters after pretraining. The parameters are averaged over 20 trials and normalized to stay in the range $[-1, 1]$.

## 7 Limitations and Future Works

While our work dives deeper into the question of why reinforcement pretraining enables ICTD, it has several limitations. Firstly, our analysis assumes a linear Transformer. Although linear attentions are usual in theoretical analysis for in-context learning [Ahn et al., 2024, Gatmiry et al., 2024, Park et al., 2025, Wang et al., 2025], it leaves a gap between theory and practice. Future works can look into extending the results to simple non-linear activations, such as ReLU. Secondly, our prompt construction assumes an enumeration of the states and expected next features. While this assumption

facilitates our analysis, it can hardly hold in practice, especially in MDPs with a continuous state space. Future works may include generalizing our results to sampled states and/or next features. Thirdly, our empirical study is relatively small in scale. While it is intended for a clear proof of concept, future work can scale up our experiments. In this work, we focus only on the prediction problem. Exciting future directions include studying the emergence of in-context control via reinforcement pretraining.

# 8 Conclusion

This work presents an in-depth analysis of the emergence of the in-context TD parameter through reinforcement pretraining. We show that under some prompt construction, the in-context prediction of the Transformer with ICTD weights converges to the true value as it gets deeper. Furthermore, we prove that the ICTD parameter is one minimizer of the NEU loss for both multi-task TD and MC pretraining as the number of attention layers goes to infinity. Through controlled experiments, we show that (1) both multi-task TD and MC induce in-context policy evaluation capabilities in linear Transformers, and (2) both pretraining schemes produce the same final parameter — the ICTD weights. We acknowledge the limitations of our work, including the linear attention assumption put on the Transformer, the practicality issue of our prompt construction, and the small scale of our empirical study. We believe addressing these limitations will make admirable contributions to the advancement of ICRL.

## Acknowledgements

This work is supported in part by the US National Science Foundation under the awards III-2128019, SLES-2331904, and CAREER-2442098, the Commonwealth Cyber Initiative's Central Virginia Node under the award VV-1Q26-001, and an NVIDIA academic grant program award.

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

# A  Multi-task TD and MC

We provide the pseudocode of multi-task TD and MC in this section.

---

**Algorithm 1:** Multi-Task Temporal Difference Learning (adapted from Algorithm 1 of Wang et al. [2025])

---

1: **Input:** state number $n$, MRP distribution $\Delta(p, r)$, one-hot encoder $\phi$, batch size $b$, number of training tasks $k$, learning rate $\alpha$, discount factor $\gamma$, transformer parameters $\theta \doteq \{P_l, Q_l\}_{l=0,1,\ldots L-1}$
2: **for** $i \leftarrow 1$ **to** $k$ **do**
3:     $p, r \sim \Delta(p, r)$
4:     $p_0 \leftarrow \texttt{stationary\_distribution}(p)$
5:     Unroll $S_0, R_1, S_1, R_2, \ldots, S_{b-1}, R_b, S_b$ from the MRP $(p_0, p, r)$
6:     $\zeta \leftarrow 0$   // to aggregate semi-gradients
7:     **for** $t = 0, \ldots, b-1$ **do**
8: 
$$Z_0 \leftarrow \begin{bmatrix} \phi(s_0) & \cdots & \phi(s_{n-1}) & \phi(S_t) \\ \gamma \sum_{j=0}^{n-1} p(s_j \mid s_0)\phi(s_j) & \cdots & \gamma \sum_{j=0}^{n-1} p(s_j \mid s_{n-1})\phi(s_j) & 0 \\ r(s_0) & \cdots & r(s_{n-1}) & 0 \end{bmatrix},$$
$$Z_0' \leftarrow \begin{bmatrix} \phi(s_0) & \cdots & \phi(s_{n-1}) & \phi(S_{t+1}) \\ \gamma \sum_{j=0}^{n-1} p(s_j \mid s_0)\phi(s_j) & \cdots & \gamma \sum_{j=0}^{n-1} p(s_j \mid s_{n-1})\phi(s_j) & 0 \\ r(s_0) & \cdots & r(s_{n-1}) & 0 \end{bmatrix},$$

9:       $\zeta \leftarrow \zeta + \alpha(R_{t+1} + \gamma \text{TF}_L(Z_0'; \theta) - \text{TF}_L(Z_0; \theta))\nabla_\theta \text{TF}_L(Z_0; \theta)$
10:    **end for**
11:    $\theta \leftarrow \theta + \frac{\zeta}{b}$   // apply update
12: **end for**

---

---

**Algorithm 2:** Multi-Task Monte Carlo Learning

---

1: **Input:** state number $n$, MRP distribution $\Delta(p, r)$, one-hot encoder $\phi$, MC unroll budget $\tau$, batch size $b$, number of training tasks $k$, learning rate $\alpha$, discount factor $\gamma$, transformer parameters $\theta \doteq \{P_l, Q_l\}_{l=0,1,\ldots L-1}$
2: **for** $i \leftarrow 1$ **to** $k$ **do**
3:     $p, r \sim \Delta(p, r)$
4:     $p_0 \leftarrow \texttt{stationary\_distribution}(p)$
5:     Unroll $S_0, R_1, S_1, R_2, \ldots, S_{b-2}, R_{b-1}, S_{b-1}$ from the MRP $(p_0, p, r)$
6:     $\zeta \leftarrow 0$   // to aggregate semi-gradients
7:     **for** $t = 0, \ldots, b-1$ **do**
8:
$$Z_0 \leftarrow \begin{bmatrix} \phi(s_0) & \cdots & \phi(s_{n-1}) & \phi(S_t) \\ \gamma \sum_{j=0}^{n-1} p(s_j \mid s_0)\phi(s_j) & \cdots & \gamma \sum_{j=0}^{n-1} p(s_j \mid s_{n-1})\phi(s_j) & 0 \\ r(s_0) & \cdots & r(s_{n-1}) & 0 \end{bmatrix}$$
      // Monte Carlo unrolling
9:       $G \leftarrow 0$
10:      $\psi \leftarrow 1$
11:      $s \leftarrow S_t$
12:      **for** $i = 1, 2, \ldots \tau$ **do**
13:        $G \leftarrow \psi r(s) + G$
14:        $\psi \leftarrow \psi \gamma$
15:        $s' \sim p(\cdot \mid s)$
16:        $s \leftarrow s'$
17:      **end for**
18:      $\zeta \leftarrow \zeta + \alpha(G - \text{TF}_L(Z_0; \theta))\nabla_\theta \text{TF}_L(Z_0; \theta)$
19:     **end for**
20:    $\theta \leftarrow \theta + \frac{\zeta}{b}$   // apply update
21: **end for**

---

# B  Proof of Theoretical Results

## B.1  Theorem 1

*Proof.* We can rewrite (5) compactly as

$$w_{l+1} = w_l + \Phi^\top r + \gamma \Phi^\top P \Phi w_l - \Phi^\top \Phi w_l$$
$$= w_l + \Phi^\top (r + (\gamma P - I) \Phi w_l).$$

Substituting $\Phi = I_n$, we have

$$w_{l+1} = w_l + r + (\gamma P - I) w_l$$
$$= r + \gamma P w_l.$$

Unrolling the recursive definition, we get

$$w_0 = 0$$
$$w_1 = r$$
$$w_2 = r + \gamma P r$$
$$w_3 = r + \gamma P r + (\gamma P)^2 r$$
$$\vdots$$
$$w_l = \sum_{k=0}^{l-1} (\gamma P)^k r. \tag{6}$$

Therefore, we have

$$\lim_{L \to \infty} w_L = \lim_{L \to \infty} \sum_{k=0}^{L-1} (\gamma P)^k r = (I - \gamma P)^{-1} r.$$

With a slight abuse of notation, we employ $v \in \mathbb{R}^n$ to denote the value function in vector form, where the $s$-th dimension is $v(s)$. When writing out $v$ as a Bellman equation, we have

$$v = r + \gamma P v$$
$$(I - \gamma P) v = r$$
$$v = (I - \gamma P)^{-1} r. \tag{7}$$

Thus, we have shown that $\lim_{L \to \infty} w_L = v$.

Finally, recall that

$$\mathrm{TF}_L(Z_0; \theta_L^{\mathrm{TD}}) = \langle \phi_q, w_L \rangle.$$

We therefore have

$$\lim_{L \to \infty} \mathrm{TF}_L(Z_0; \theta_L^{\mathrm{TD}}) = \lim_{L \to \infty} \langle \phi_q, w_L \rangle = \left\langle \phi_q, \lim_{L \to \infty} w_L \right\rangle = \langle \phi_q, v \rangle = v(s_q).$$

$\square$

## B.2  Lemma 1

*Proof.*

$$\left| \mathbb{E}_{s_q' \sim p(s_q)} \left[ r(s_q) + \gamma \mathrm{TF}_L(Z_0'; \theta_L^{\mathrm{TD}}) - \mathrm{TF}_L(Z_0; \theta_L^{\mathrm{TD}}) \right] \right|$$
$$= \left| r(s_q) + \gamma \mathbb{E}_{s_q' \sim p(s_q)} \left[ \mathrm{TF}_L(Z_0'; \theta_L^{\mathrm{TD}}) \right] - \mathrm{TF}_L(Z_0; \theta_L^{\mathrm{TD}}) \right|$$
$$= \left| \langle \phi_q, r \rangle + \gamma \mathbb{E}_{s_q' \sim p(s_q)} \left[ \langle \phi_q', w_L \rangle \right] - \langle \phi_q, w_L \rangle \right|$$
$$= \left| \phi_q^\top r + \gamma \mathbb{E}_{s_q' \sim p(s_q)} [\phi_q']^\top \sum_{k=0}^{L-1} (\gamma P)^k r - \phi_q^\top \sum_{k=0}^{L-1} (\gamma P)^k r \right| \qquad \text{(eq. (6))}$$

$$
\begin{aligned}
&= \left| \phi_q^\top r + \gamma \big( \phi_q^\top P \big) \sum_{k=0}^{L-1} (\gamma P)^k r - \phi_q^\top \sum_{k=0}^{L-1} (\gamma P)^k r \right| \\
&= \left| \phi_q^\top r + \phi_q^\top \sum_{k=1}^{L} (\gamma P)^k r - \phi_q^\top \sum_{k=0}^{L-1} (\gamma P)^k r \right| \\
&= \left| \phi_q^\top r + \phi_q^\top (\gamma P)^L r - \phi_q^\top r \right| \\
&= \gamma^L \big| \langle \phi_q, P^L r \rangle \big| \\
&\le \gamma^L \|\phi_q\|_1 \|P^L r\|_\infty && \text{(Hölder's inequality)} \\
&= \gamma^L \|P^L r\|_\infty && (\|\phi_q\|_1 = 1) \\
&\le \gamma^L \|P^L\|_\infty \|r\|_\infty \\
&= \gamma^L \|r\|_\infty && (\|P^L\|_\infty = 1)
\end{aligned}
$$

$\square$

### B.3   Lemma 2

*Proof.* We first note that $y_l^q = -\mathrm{TF}_L(Z_0; \theta_L^{\mathrm{TD}}) = -\langle \phi_q, w_l \rangle$. Therefore, we get

$$
\begin{aligned}
|y_l^q| &= |\langle \phi_q, w_l \rangle| \\
&\le \|\phi_q\|_1 \|w_l\|_\infty && \text{(Hölder's inequality)} \\
&= \|w_l\|_\infty && (\|\phi_q\|_1 = 1) \\
&= \left\| \sum_{k=0}^{l-1} (\gamma P)^k r \right\|_\infty && \text{(eq. (6))} \\
&= \left\| \big( I - (\gamma P)^l \big) (I - \gamma P)^{-1} r \right\|_\infty \\
&= \left\| \big( I - (\gamma P)^l \big) v \right\|_\infty && \text{(eq. (7))} \\
&= \left\| v - \gamma^l P^l v \right\|_\infty \\
&\le \|v\|_\infty + \|\gamma^l P^l v\|_\infty \\
&\le \|v\|_\infty + \gamma^l \|P^l\|_\infty \|v\|_\infty \\
&= (1 + \gamma^l) \|v\|_\infty && (\|P^l\|_\infty = 1)
\end{aligned}
$$

We have proved 1. of Lemma 2.

Under the structure of $P_l$ in $\theta_L^{\mathrm{TD}}$, it holds that the first $2n$ rows of $Z_l$ remain unchanged. Thus, we drop the subscript of $X_l$ and write

$$
Z_l = \begin{bmatrix} X & x^q \\ Y_l & y_l^q \end{bmatrix}.
$$

We then proceed to compute the recursive forms of $Y_l$ and $y_l^q$. We first have

$$
\begin{aligned}
P_l Z_l M &= \begin{bmatrix} 0_{2n \times 2n} & 0_{2n \times 1} \\ 0_{1 \times 2n} & 1 \end{bmatrix} \begin{bmatrix} X & x^q \\ Y_l & y_l^q \end{bmatrix} \begin{bmatrix} I_n & 0_{n \times 1} \\ 0_{1 \times n} & 0 \end{bmatrix} \\
&= \begin{bmatrix} 0_{2n \times 2n} & 0_{2n \times 1} \\ 0_{1 \times 2n} & 1 \end{bmatrix} \begin{bmatrix} X & 0_{2n \times 1} \\ Y_l & 0 \end{bmatrix} \\
&= \begin{bmatrix} 0_{2n \times n} & 0_{2n \times 1} \\ Y_l & 0 \end{bmatrix}.
\end{aligned}
$$

Next, we have

$$
\begin{aligned}
Z_l^\top Q_l Z_l &= \begin{bmatrix} X^\top & Y_l^\top \\ x^{q\top} & y_l^q \end{bmatrix} \begin{bmatrix} A_l & 0_{2n \times 1} \\ 0_{1 \times 2n} & 0 \end{bmatrix} \begin{bmatrix} X & x^q \\ Y_l & y_l^q \end{bmatrix} \\
&= \begin{bmatrix} X^\top & Y_l^\top \\ x^{q\top} & y_l^q \end{bmatrix} \begin{bmatrix} A_l X & A_l x^q \\ 0_{1 \times n} & 0 \end{bmatrix}
\end{aligned}
$$

$$= \begin{bmatrix} X^\top A_l X & X^\top A_l x^q \\ x^{q\top} A_l X & x^{q\top} A_l x^q \end{bmatrix}.$$

Putting the two parts together, we have

$$P_l Z_l M (Z_l^\top Q_l Z_l) = \begin{bmatrix} 0_{2n \times n} & 0_{2n \times 1} \\ Y_l & 0 \end{bmatrix} \begin{bmatrix} X^\top A_l X & X^\top A_l x^q \\ x^{q\top} A_l X & x^{q\top} A_l x^q \end{bmatrix}$$

$$= \begin{bmatrix} 0_{2n \times n} & 0_{2n \times 1} \\ Y_l X^\top A_l X & Y_l X^\top A_l x^q \end{bmatrix}.$$

We therefore get

$$\begin{cases} Y_{l+1} & = Y_l + Y_l X^\top A_l X; \\ y_{l+1}^q & = y_l^q + Y_l X^\top A_l x^q. \end{cases}$$

By induction, we can prove that

$$Y_{l+1} = Y_0 + \sum_{i=0}^{l} Y_i X^\top A_i X.$$

Similarly, we can show that

$$y_{l+1}^q = y_0^q + \sum_{i=0}^{l} Y_i X^\top A_i x^q.$$

Since $y_0^q = 0$, we get

$$y_{l+1}^q = \sum_{i=0}^{l} Y_i X^\top A_i x^q. \tag{8}$$

Let $y_l^{(k)}$ denote the $k$-th component of $Y_l$. It is clear that

$$y_{l+1}^{(k)} = y_0^{(k)} + \sum_{i=0}^{l} Y_i X^\top A_i x^{(k)}, \tag{9}$$

where $x^{(k)}$ denotes the $k$-th column of $X$. Recall that $y_l^q = -\langle \phi_q, w_l \rangle$. Substituting $x^{(k)}$ in (9) into $x^q$ in (8), we get

$$y_l^{(k)} = y_0^{(k)} - \langle \phi_k, w_l \rangle.$$

Therefore, we have

$$Y_l = Y_0 - w_l^\top I_n = r^\top - w_l^\top.$$

We finally get

$$\begin{aligned} \|Y_l\|_1 &= \left\| Y_l^\top \right\|_\infty \\ &= \|r - w_l\|_\infty \\ &= \left\| r - \sum_{k=0}^{l-1} (\gamma P)^k r \right\|_\infty \\ &= \left\| r - \left( I - (\gamma P)^l \right) (I - \gamma P)^{-1} r \right\|_\infty \\ &= \left\| r - \left( I - (\gamma P)^l \right) v \right\|_\infty & \text{(eq. (7))} \\ &= \left\| (I - \gamma P)(I - \gamma P)^{-1} r - \left( I - (\gamma P)^l \right) v \right\|_\infty \\ &= \left\| (I - \gamma P) v - \left( I - (\gamma P)^l \right) v \right\|_\infty \\ &= \left\| (\gamma P)^l v - \gamma P v \right\|_\infty \\ &\leq \left\| (\gamma P)^l v \right\|_\infty + \|\gamma P v\|_\infty \\ &\leq \gamma^l \left\| P^l \right\|_\infty \|v\|_\infty + \gamma \|P\|_\infty \|v\|_\infty \\ &= (\gamma^l + \gamma) \|v\|_\infty. & (\left\| P^l \right\|_\infty = 1, \|P\|_\infty = 1) \end{aligned}$$

This concludes the proof of 2. of Lemma 2. $\qquad \square$

## B.4 Lemma 3

*Proof.* Since $\theta_L^{\text{TD}} \in \Theta^{\text{Looped}}$, we can drop the subscripts of the linear Transformer parameters because of parameter sharing across the layers. We further have $\nabla_\theta \text{TF}_L\left(Z_0; \theta_L^{\text{TD}}\right) \doteq \begin{bmatrix} \text{vec}\left(\nabla_A \text{TF}_L\left(Z_0; \theta_L^{\text{TD}}\right)\right) \\ \nabla_u^\top \text{TF}_L\left(Z_0; \theta_L^{\text{TD}}\right) \end{bmatrix} \in \mathbb{R}^{(2n+1)(2n)}$, where vec denotes the vectorization operation. We then have

$$\begin{cases} Y_{l+1} &= Y_l + (uX + Y_l)X^\top AX = uXX^\top AX + Y_l\left(I + X^\top AX\right); \\ y_{l+1}^q &= y_l^q + (uX + Y_l)X^\top Ax^q = y_l^q + uXX^\top Ax^q + Y_lX^\top Ax^q. \end{cases} \tag{10}$$

We begin by solving for $\nabla_A y_{l+1}^{(i)}$, where $y_{l+1}^{(i)}$ denotes the $i$-th component of $Y_{l+1}$. Define $f(A) \doteq I + X^\top AX$ to simplify notations. By (10), we have

$$\begin{aligned} \nabla_A y_{l+1}^{(i)} &= \frac{\mathrm{d}\left[uXX^\top AX\right]^{(i)}}{\mathrm{d}A} + \frac{\mathrm{d}[Y_l f(A)]^{(i)}}{\mathrm{d}A} \\ &= \frac{\mathrm{d}\left[uXX^\top Ax^{(i)}\right]}{\mathrm{d}A} + \frac{\mathrm{d}\left[\sum_{j=1}^n y_l^{(j)} f(A)_{j,i}\right]}{\mathrm{d}A}, \end{aligned}$$

where $x^{(i)}$ indicates the $i$-th column of $X$. We analyze the summands separately. Since $\nabla_M\left(a^\top Mb\right) = ab^\top$ given matrix $M$ and vectors $a$ and $b$, it holds that

$$\frac{\mathrm{d}\left[uXX^\top Ax^{(i)}\right]}{\mathrm{d}A} = XX^\top u^\top x^{(i)\top}.$$

Plugging in the condition that $u = 0$ in $\theta_L^{\text{TD}}$, we get $\frac{\mathrm{d}\left[uXX^\top Ax^{(i)}\right]}{\mathrm{d}A} = 0$.

Next, we tackle the gradient term $\frac{\mathrm{d}\left[\sum_{j=1}^n y_l^{(j)} f(A)_{j,i}\right]}{\mathrm{d}A}$. By the summation rule and the product rule, we get

$$\frac{\mathrm{d}\left[\sum_{j=1}^n y_l^{(j)} f(A)_{j,i}\right]}{\mathrm{d}A} = \sum_{j=1}^n \frac{\mathrm{d}\left[y_l^{(j)} f(A)_{j,i}\right]}{\mathrm{d}A} = \sum_{j=1}^n \left(f(A)_{j,i} \frac{\mathrm{d}y_l^{(j)}}{\mathrm{d}A} + y_l^{(j)} \frac{\mathrm{d}f(A)_{j,i}}{\mathrm{d}A}\right). \tag{11}$$

Employing the structure of $A$, we first note that

$$\begin{aligned} f(A) &= I + X^\top AX \\ &= I + [I \quad \gamma P] \begin{bmatrix} -I & I \\ 0 & 0 \end{bmatrix} \begin{bmatrix} I \\ \gamma P^\top \end{bmatrix} \\ &= I + [-I \quad I] \begin{bmatrix} I \\ \gamma P^\top \end{bmatrix} \\ &= I + \gamma P^\top - I \\ &= \gamma P^\top. \end{aligned}$$

We also note that

$$\frac{\mathrm{d}f(A)_{j,i}}{\mathrm{d}A} = \frac{\mathrm{d}\left[I_{j,i} + x^{(j)\top} Ax^{(i)}\right]}{\mathrm{d}A} = x^{(j)} x^{(i)\top}.$$

Therefore, plugging them in (11), we have

$$\begin{aligned} \frac{\mathrm{d}\left[\sum_{j=1}^n y_l^{(j)} f(A)_{j,i}\right]}{\mathrm{d}A} &= \sum_{j=1}^n \left(\gamma P_{i,j} \nabla_A y_l^{(j)} + y_l^{(j)} x^{(j)} x^{(i)\top}\right) \\ &= XY_l^\top x^{(i)\top} + \sum_{j=1}^n \gamma P_{i,j} \nabla_A y_l^{(j)}. \end{aligned}$$

Consequently, we have

$$\nabla_A y_{l+1}^{(i)} = XY_l^\top x^{(i)\top} + \sum_{j=1}^n \gamma P_{i,j} \nabla_A y_l^{(j)}.$$

We now bound the gradient term norm $\left\|\nabla_A y_{l+1}^{(i)}\right\|_1$. We first have

$$\begin{aligned}
\left\|\nabla_A y_{l+1}^{(i)}\right\|_1 &= \left\|XY_l^\top x^{(i)\top} + \sum_{j=1}^n \gamma P_{i,j} \nabla_A y_l^{(j)}\right\|_1 \\
&\le \left\|XY_l^\top x^{(i)\top}\right\|_1 + \left\|\sum_{j=1}^n \gamma P_{i,j} \nabla_A y_l^{(j)}\right\|_1 \\
&\le \|X\|_1 \|Y_l^\top\|_1 \left\|x^{(i)\top}\right\|_1 + \gamma \sum_{j=1}^n P_{i,j} \left\|\nabla_A y_l^{(j)}\right\|_1 \\
&= \left\|\begin{bmatrix} I \\ \gamma P^\top \end{bmatrix}\right\|_1 \|Y_l\|_\infty \left\|x^{(i)}\right\|_\infty + \gamma \sum_{j=1}^n P_{i,j} \left\|\nabla_A y_l^{(j)}\right\|_1 \\
&= (1+\gamma)\|Y_l\|_\infty + \gamma \sum_{j=1}^n P_{i,j} \left\|\nabla_A y_l^{(j)}\right\|_1 \\
&\le n(1+\gamma)(\gamma^l + \gamma)\|v\|_\infty + \gamma \sum_{j=1}^n P_{i,j} \left\|\nabla_A y_l^{(j)}\right\|_1 \qquad \text{(Lemma 2)} \\
&\le n(1+\gamma)^2\|v\|_\infty + \gamma \sum_{j=1}^n P_{i,j} \left\|\nabla_A y_l^{(j)}\right\|_1.
\end{aligned}$$

Let $\beta \doteq n(1+\gamma)^2\|v\|_\infty$. Having observed this pattern, we claim

$$\left\|\nabla_A y_l^{(i)}\right\|_1 \le l\beta \tag{12}$$

for all $i = 1, 2, \dots, n$ and prove it by induction. Since $Y_0$ is not dependent on $A$, we have $\nabla_A Y_0 = 0$. Therefore, the base case $\left\|\nabla_A y_0^{(i)}\right\|_1 \le 0$ trivially holds. Then, suppose (12) holds for $l$. We have

$$\begin{aligned}
\left\|\nabla_A y_{l+1}^{(i)}\right\|_1 &\le \beta + \gamma \sum_{j=1}^n P_{i,j} \left\|\nabla_A y_l^{(j)}\right\|_1 \\
&\le \beta + \gamma \sum_{j=1}^n P_{i,j} l\beta \\
&= \beta + \gamma l\beta \\
&= (1 + \gamma l)\beta \\
&< (1 + l)\beta.
\end{aligned}$$

The bound in (12) is proved.

We then proceed with $\nabla_A y_{l+1}^q$. Define $g(A) \doteq X^\top A x^q$ to simplify notations. Similar to $\nabla_A Y_l$, by (10), we have

$$\begin{aligned}
\nabla_A y_{l+1}^q &= \nabla_A y_l^q + \frac{\mathrm{d}\left[uXX^\top A x^q\right]}{\mathrm{d}A} + \frac{\mathrm{d}[Y_l g(A)]}{\mathrm{d}A} \\
&= \nabla_A y_l^q + \frac{\mathrm{d}\left[uXX^\top A x^q\right]}{\mathrm{d}A} + \frac{\mathrm{d}\left[\sum_{k=1}^n y_l^{(k)} g(A)_k\right]}{\mathrm{d}A}.
\end{aligned}$$

Firstly, we get

$$\frac{\mathrm{d}\left[uXX^\top Ax^q\right]}{\mathrm{d}A} = XX^\top u^\top x^{q^\top}.$$

Substituting $u = 0$ into the equation, we get $\frac{\mathrm{d}\left[uXX^\top Ax^q\right]}{\mathrm{d}A} = 0$. Then, we have

$$\frac{\mathrm{d}\left[\sum_{k=1}^n y_l^{(k)} g(A)_k\right]}{\mathrm{d}A} = \sum_{k=1}^n \frac{\mathrm{d}\left[y_l^{(k)} g(A)_k\right]}{\mathrm{d}A} = \sum_{k=1}^n \left(g(A)_k \frac{\mathrm{d}y_l^{(k)}}{\mathrm{d}A} + y_l^{(k)} \frac{\mathrm{d}g(A)_k}{\mathrm{d}A}\right) \quad (13)$$

by the summation rule and the product rule. We note that, by the structure of $A$ in $\theta_L^{\mathrm{TD}}$, we have

$$\begin{aligned}
g(A) &= X^\top A x^q \\
&= [I \quad \gamma P]\begin{bmatrix} -I & I \\ 0 & 0 \end{bmatrix}\begin{bmatrix} \phi_q \\ 0 \end{bmatrix} \\
&= [I \quad \gamma P]\begin{bmatrix} -\phi_q \\ 0 \end{bmatrix} \\
&= -\phi_q. \quad (14)
\end{aligned}$$

We further note that

$$\frac{\mathrm{d}g(A)_k}{\mathrm{d}A} = \frac{\mathrm{d}\left[x^{(k)^\top} Ax^q\right]}{\mathrm{d}A} = x^{(k)} x^{q^\top}.$$

Combining these and plugging into (13), we get

$$\begin{aligned}
\frac{\mathrm{d}\left[\sum_{k=1}^n y_l^{(k)} g(A)_k\right]}{\mathrm{d}A} &= \sum_{k=1}^n \left(y_l^{(k)} x^{(k)} x^{q^\top} - [\phi_q]_k \nabla_A y_l^{(k)}\right) \\
&= XY_l^\top x^{q^\top} - \sum_{k=1}^n [\phi_q]_k \nabla_A y_l^{(k)}.
\end{aligned}$$

Since $\phi_q$ is a one-hot vector, we let $\omega$ be the index where $[\phi_q]_\omega = 1$. We then have

$$\frac{\mathrm{d}\left[\sum_{k=1}^n y_l^{(k)} g(A)_k\right]}{\mathrm{d}A} = XY_l^\top x^{q^\top} - \nabla_A y_l^{(\omega)}.$$

Hence, we have

$$\nabla_A y_{l+1}^q = \nabla_A y_l^q + XY_l^\top x^{q^\top} - \nabla_A y_l^{(\omega)}.$$

Now, we derive the bound for $\left\|\nabla_A y_{l+1}^q\right\|_1$. We first have

$$\begin{aligned}
\left\|\nabla_A y_{l+1}^q\right\|_1 &= \left\|\nabla_A y_l^q + XY_l^\top x^{q^\top} - \nabla_A y_l^{(\omega)}\right\|_1 \\
&\leq \left\|\nabla_A y_l^q\right\|_1 + \left\|XY_l^\top x^{q^\top}\right\|_1 + \left\|\nabla_A y_l^{(\omega)}\right\|_1.
\end{aligned}$$

We note that

$$\begin{aligned}
\left\|XY_l^\top x^{q^\top}\right\|_1 &\leq \left\|\begin{bmatrix} I \\ \gamma P^\top \end{bmatrix}\right\|_1 \left\|Y_l^\top\right\|_1 \left\|x^{q^\top}\right\|_1 \\
&= (1+\gamma)\|Y_l\|_\infty \\
&\leq n(1+\gamma)(\gamma^l + \gamma)\|v\|_\infty \quad \text{(Lemma 2)} \\
&\leq n(1+\gamma)^2 \|v\|_\infty = \beta.
\end{aligned}$$

Therefore, by (12), we have

$$\left\|\nabla_A y_{l+1}^q\right\|_1 \leq \left\|\nabla_A y_l^q\right\|_1 + (l+1)\beta.$$

With this form, we claim and prove by induction that

$$\left\|\nabla_A y_l^q\right\|_1 \leq \frac{l^2 + l}{2}\beta \tag{15}$$

for $l \geq 0$. Since $y_0^q$ does not depend on $A$, we have $\nabla_A y_0^q = 0$. Therefore, the base case $\left\|\nabla_A y_0^q\right\|_1 \leq 0$ trivially holds. Now suppose (15) holds for $l$. We get

$$\begin{aligned}
\left\|\nabla_A y_{l+1}^q\right\|_1 &\leq \left\|\nabla_A y_l^q\right\|_1 + (l+1)\beta \\
&\leq \frac{l^2 + l}{2}\beta + (l+1)\beta \\
&= \frac{(l+1)^2 + (l+1)}{2}\beta.
\end{aligned}$$

The bound in (15) is proved.

Next, we solve for $\nabla_u Y_{l+1}$. By (10), we first have

$$\begin{aligned}
\nabla_u y_{l+1}^{(i)} &= \frac{d\left[uXX^\top A x^{(i)}\right]}{du} + \frac{d\left[\gamma \sum_{j=1}^n P_{i,j} y_l^{(j)}\right]}{du} \\
&= XX^\top A x^{(i)} + \gamma \sum_{j=1}^n P_{i,j} \nabla_u y_l^{(j)}.
\end{aligned}$$

Define $\nabla_u Y_l \doteq \begin{bmatrix} \nabla_u y_l^{(1)} & \nabla_u y_l^{(2)} & \cdots & \nabla_u y_l^{(n)} \end{bmatrix} \in \mathbb{R}^{2n \times n}$. We thus have

$$\nabla_u Y_{l+1} = XX^\top A X + \gamma \nabla_u Y_l P^\top.$$

We now bound $\left\|\nabla_u Y_{l+1}\right\|_1$. We get

$$\begin{aligned}
\left\|\nabla_u Y_{l+1}\right\|_1 &= \left\|XX^\top A X + \gamma \nabla_u Y_l P^\top\right\|_1 \\
&\leq \left\|XX^\top A X\right\|_1 + \left\|\gamma \nabla_u Y_l P^\top\right\|_1 \\
&\leq \|X\|_1 \left\|X^\top A X\right\|_1 + \gamma \|\nabla_u Y_l\|_1 \left\|P^\top\right\|_1 \\
&= \left\|\begin{bmatrix} I \\ \gamma P^\top \end{bmatrix}\right\|_1 \left\|[I \quad \gamma P]\begin{bmatrix} -I & I \\ 0 & 0 \end{bmatrix}\begin{bmatrix} I \\ \gamma P^\top \end{bmatrix}\right\|_1 + \gamma \|\nabla_u Y_l\|_1 \\
&= (1+\gamma)\left\|\gamma P^\top - I\right\|_1 + \gamma \|\nabla_u Y_l\|_1 \\
&\leq (1+\gamma)^2 + \gamma \|\nabla_u Y_l\|_1.
\end{aligned}$$

Let $\zeta \doteq (1+\gamma)^2$. We claim

$$\|\nabla_u Y_l\|_1 \leq l\zeta \tag{16}$$

and prove it by induction. We first have $\nabla_u Y_0 = 0$ because $Y_0$ does not depend on $u$. Hence, the base case $\|\nabla_u Y_0\|_1 \leq 0$ holds trivially. Assume (16) holds for $l$. We have

$$\begin{aligned}
\|\nabla_u Y_{l+1}\|_1 &\leq \zeta + \gamma \|\nabla_u Y_l\|_1 \\
&\leq \zeta + \gamma l \zeta \\
&\leq (1 + \gamma l)\zeta \\
&< (1 + l)\zeta.
\end{aligned}$$

The bound in (16) holds.

Finally, we compute $\nabla_u y_{l+1}^q$. By (10), we have

$$\begin{aligned}
\nabla_u y_{l+1}^q &= \nabla_u y_l^q + \frac{d\left[uXX^\top A x^q\right]}{du} + \frac{d\left[Y_l X^\top A x^q\right]}{du} \\
&= \nabla_u y_l^q + XX^\top A x^q + (\nabla_u Y_l)X^\top A x^q.
\end{aligned}$$

Then, we bound $\left\|\nabla_u y_{l+1}^q\right\|_1$ by

$$\left\|\nabla_u y_{l+1}^q\right\|_1 = \left\|\nabla_u y_l^q + XX^\top A x^q + (\nabla_u Y_l)X^\top A x^q\right\|_1$$

$$\leq \|\nabla_u y_l^q\|_1 + \|XX^\top Ax^q\|_1 + \|(\nabla_u Y_l)X^\top Ax^q\|_1$$
$$\leq \|\nabla_u y_l^q\|_1 + \|X\|_1\|g(A)\|_1 + \|(\nabla_u Y_l)\|_1\|g(A)\|_1$$
$$\leq \|\nabla_u y_l^q\|_1 + (1+\gamma)\|g(A)\|_1 + l\zeta\|g(A)\|_1 \qquad \text{(eq. (16))}$$
$$= \|\nabla_u y_l^q\|_1 + (1+\gamma)\|\phi_q\|_1 + l\zeta\|\phi_q\|_1 \qquad \text{(eq. (14))}$$
$$= \|\nabla_u y_l^q\|_1 + (1+\gamma) + l\zeta$$

Define $\eta \doteq (1+\gamma)$. We prove the bound

$$\|\nabla_u y_l^q\|_1 \leq \frac{l^2+l}{2}\zeta + l\eta \qquad (17)$$

by induction. We first have the base case $\|\nabla_u y_0^q\|_1 \leq 0$ holds because $y_0^q$ is independent from $u$ and thus $\nabla_u y_0^q = 0$. Now, suppose (17) holds for $l$. We have

$$\begin{aligned}
\|\nabla_u y_{l+1}^q\|_1 &\leq \|\nabla_u y_l^q\|_1 + \eta + l\zeta \\
&\leq \frac{l^2+l}{2}\zeta + l\eta + \eta + l\zeta \\
&\leq \frac{l^2+l}{2}\zeta + (l+1)\zeta + (l+1)\eta \\
&= \frac{l^2+3l+2}{2}\zeta + (l+1)\eta \\
&= \frac{(l+1)^2+(l+1)}{2}\zeta + (l+1)\eta.
\end{aligned}$$

The bound holds for (17).

Since $\text{TF}_L(Z_0; \theta_L^{\text{TD}}) = -y_L^q$, we have

$$\begin{cases}
\|\nabla_A \text{TF}_L(Z_0; \theta_L^{\text{TD}})\|_1 = \|\nabla_A y_L^q\|_1 \leq \frac{L^2+L}{2}\beta; \\
\|\nabla_u \text{TF}_L(Z_0; \theta_L^{\text{TD}})\|_1 = \|\nabla_u y_L^q\|_1 \leq \frac{L^2+L}{2}\zeta + L\eta.
\end{cases}$$

We then have bounds

$$\begin{cases}
\|\text{vec}(\nabla_A \text{TF}_L(Z_0; \theta_L^{\text{TD}}))\|_1 \leq 2n\left(\frac{L^2+L}{2}\beta\right); \\
\|\nabla_u^\top \text{TF}_L(Z_0; \theta_L^{\text{TD}})\|_1 \leq 2n\left(\frac{L^2+L}{2}\zeta + L\eta\right).
\end{cases}$$

We therefore get

$$\begin{aligned}
\|\nabla_\theta \text{TF}_L(Z_0; \theta_L^{\text{TD}})\|_1 &= \left\|\begin{bmatrix} \text{vec}(\nabla_A \text{TF}_L(Z_0; \theta_L^{\text{TD}})) \\ \nabla_u^\top \text{TF}_L(Z_0; \theta_L^{\text{TD}}) \end{bmatrix}\right\|_1 \\
&= \|\text{vec}(\nabla_A \text{TF}_L(Z_0; \theta_L^{\text{TD}}))\|_1 + \|\nabla_u^\top \text{TF}_L(Z_0; \theta_L^{\text{TD}})\|_1 \\
&\leq \frac{L^2+L}{2}(2n(\beta+\zeta)) + L(2n\eta).
\end{aligned}$$

Let $\nu \doteq 2n(\beta+\zeta)$ and $\xi \doteq 2n\eta$. We finally arrive at

$$\|\nabla_\theta \text{TF}_L(Z_0; \theta_L^{\text{TD}})\|_1 \leq \frac{L^2+L}{2}\nu + L\xi.$$

$\square$

## B.5  Theorem 2

*Proof.* We have

$$J(\theta_L^{\text{TD}})$$

$$\doteq \left\|\mathbb{E}_{s_q\sim\mu^p,s_q'\sim p(s_q),p\sim\Delta(p),r\sim\Delta(r)}\left[\left(r(s_q)+\gamma\mathrm{TF}_L\left(Z_0';\theta_L^{\mathrm{TD}}\right)-\mathrm{TF}_L\left(Z_0;\theta_L^{\mathrm{TD}}\right)\right)\nabla_\theta\mathrm{TF}_L\left(Z_0;\theta_L^{\mathrm{TD}}\right)\right]\right\|_1$$

$$=\left\|\mathbb{E}_{s_q\sim\mu^p,p\sim\Delta(p),r\sim\Delta(r)}\left[\mathbb{E}_{s_q'\sim p(s_q)}\left[\left(r(s_q)+\gamma\mathrm{TF}_L\left(Z_0';\theta_L^{\mathrm{TD}}\right)-\mathrm{TF}_L\left(Z_0;\theta_L^{\mathrm{TD}}\right)\right)\nabla_\theta\mathrm{TF}_L\left(Z_0;\theta_L^{\mathrm{TD}}\right)\right]\right]\right\|_1$$

$$=\left\|\mathbb{E}_{s_q\sim\mu^p,p\sim\Delta(p),r\sim\Delta(r)}\left[\mathbb{E}_{s_q'\sim p(s_q)}\left[\left(r(s_q)+\gamma\mathrm{TF}_L\left(Z_0';\theta_L^{\mathrm{TD}}\right)-\mathrm{TF}_L\left(Z_0;\theta_L^{\mathrm{TD}}\right)\right)\right]\nabla_\theta\mathrm{TF}_L\left(Z_0;\theta_L^{\mathrm{TD}}\right)\right]\right\|_1$$

$$\leq\mathbb{E}_{s_q\sim\mu^p,p\sim\Delta(p),r\sim\Delta(r)}\left[\left\|\mathbb{E}_{s_q'\sim p(s_q)}\left[\left(r(s_q)+\gamma\mathrm{TF}_L\left(Z_0';\theta_L^{\mathrm{TD}}\right)-\mathrm{TF}_L\left(Z_0;\theta_L^{\mathrm{TD}}\right)\right)\right]\nabla_\theta\mathrm{TF}_L\left(Z_0;\theta_L^{\mathrm{TD}}\right)\right\|_1\right]$$

$$=\mathbb{E}_{s_q\sim\mu^p,p\sim\Delta(p),r\sim\Delta(r)}\left[\left|\mathbb{E}_{s_q'\sim p(s_q)}\left[\left(r(s_q)+\gamma\mathrm{TF}_L\left(Z_0';\theta_L^{\mathrm{TD}}\right)-\mathrm{TF}_L\left(Z_0;\theta_L^{\mathrm{TD}}\right)\right)\right]\right|\left\|\nabla_\theta\mathrm{TF}_L\left(Z_0;\theta_L^{\mathrm{TD}}\right)\right\|_1\right]$$

$$\leq\mathbb{E}_{s_q\sim\mu^p,p\sim\Delta(p),r\sim\Delta(r)}\left[\|r\|_\infty\gamma^L\left\|\nabla_{\theta_L}\mathrm{TF}_L\left(Z_0;\theta_L^{\mathrm{TD}}\right)\right\|_1\right] \qquad\text{(Lemma 1)}$$

$$\leq\mathbb{E}_{s_q\sim\mu^p,p\sim\Delta(p),r\sim\Delta(r)}\left[\|r\|_\infty\gamma^L\left(\frac{L^2+L}{2}\nu+L\xi\right)\right] \qquad\text{(Lemma 3)}$$

$$=\mathbb{E}_{r\sim\Delta(r)}[\|r\|_\infty]\gamma^L\left(\frac{L^2+L}{2}\nu+L\xi\right).$$

Let $C_r\doteq\mathbb{E}_{r\sim\Delta(r)}[\|r\|_\infty]$. We have $J(\theta_L^{\mathrm{TD}})\leq\frac{L^2+L}{2}\nu C_r\gamma^L+L\xi C_r\gamma^L$. Therefore, it holds that

$$\lim_{L\to\infty}J(\theta_L^{\mathrm{TD}})\leq\lim_{L\to\infty}\frac{L^2+L}{2}\nu C_r\gamma^L+L\xi C_r\gamma^L=0$$

because the exponential term $\gamma^L$ dominates the polynomial terms of $L$. Since $J(\theta_L^{\mathrm{TD}})$ is a norm and thus always nonnegative, we have

$$\lim_{L\to\infty}J(\theta_L^{\mathrm{TD}})=0,$$

completing the proof. $\qquad\square$

### B.6 Lemma 4

*Proof.* We note that given $p,r,s_q$, there is no randomness in $\mathrm{TF}_L(Z_0;\theta)$. Therefore, we have

$$\left|\mathbb{E}_{p,r}\left[G(s_q)-\mathrm{TF}_L\left(Z_0;\theta_L^{\mathrm{TD}}\right)\right]\right|$$
$$=\left|\mathbb{E}_{p,r}[G(s_q)]-\mathrm{TF}_L\left(Z_0;\theta_L^{\mathrm{TD}}\right)\right|$$
$$=|v(s_q)-\langle\phi_q,w_L\rangle|$$
$$=\left|v(s_q)-\phi_q^\top\sum_{k=0}^{L-1}(\gamma P)^k r\right| \qquad\text{(eq. (6))}$$
$$=\left|\phi_q^\top v-\phi_q^\top\left(I-(\gamma P)^L\right)(I-\gamma P)^{-1}r\right|$$
$$=\left|\phi_q^\top v-\phi_q^\top\left(I-\gamma^L P^L\right)v\right|$$
$$=\gamma^L\left|\phi_q^\top P^L v\right|$$
$$\leq\gamma^L\|\phi_q\|_1\|P^L v\|_\infty \qquad\text{(Hölder's inequality)}$$
$$\leq\gamma^L\|P^L\|_\infty\|v\|_\infty \qquad(\|\phi_q\|_1=1)$$
$$=\gamma^L\|v\|_\infty. \qquad(\|P^L\|_\infty=1)$$
$$\square$$

### B.7 Corollary 1

*Proof.* (Corollary 1)
$$J'(\theta_L^{\mathrm{TD}})$$

$$=\left\|\mathbb{E}_{s_q\sim\mu^p,p\sim\Delta(p),r\sim\Delta(r)}\left[\left(G(s_q)-\mathrm{TF}_L\left(Z_0;\theta_L^{\mathrm{TD}}\right)\right)\nabla_\theta\mathrm{TF}_L\left(Z_0;\theta_L^{\mathrm{TD}}\right)\right]\right\|_1$$

$$=\left\|\mathbb{E}_{s_q\sim\mu^p,p\sim\Delta(p),r\sim\Delta(r)}\left[\mathbb{E}_{p,r}\left[\left(G(s_q)-\mathrm{TF}_L\left(Z_0;\theta_L^{\mathrm{TD}}\right)\right)\nabla_\theta\mathrm{TF}_L\left(Z_0;\theta_L^{\mathrm{TD}}\right)\right]\right]\right\|_1$$

$$=\left\|\mathbb{E}_{s_q\sim\mu^p,p\sim\Delta(p),r\sim\Delta(r)}\left[\mathbb{E}_{p,r}\left[G(s_q)-\mathrm{TF}_L\left(Z_0;\theta_L^{\mathrm{TD}}\right)\right]\nabla_\theta\mathrm{TF}_L\left(Z_0;\theta_L^{\mathrm{TD}}\right)\right]\right\|_1$$

$$\leq\mathbb{E}_{s_q\sim\mu^p,p\sim\Delta(p),r\sim\Delta(r)}\left[\left\|\mathbb{E}_{p,r}\left[G(s_q)-\mathrm{TF}_L\left(Z_0;\theta_L^{\mathrm{TD}}\right)\right]\nabla_\theta\mathrm{TF}_L\left(Z_0;\theta_L^{\mathrm{TD}}\right)\right\|_1\right]$$

$$=\mathbb{E}_{s_q\sim\mu^p,p\sim\Delta(p),r\sim\Delta(r)}\left[\left|\mathbb{E}_{p,r}\left[G(s_q)-\mathrm{TF}_L\left(Z_0;\theta_L^{\mathrm{TD}}\right)\right]\right|\left\|\nabla_\theta\mathrm{TF}_L\left(Z_0;\theta_L^{\mathrm{TD}}\right)\right\|_1\right]$$

$$\leq\mathbb{E}_{s_q\sim\mu^p,p\sim\Delta(p),r\sim\Delta(r)}\left[\|v\|_\infty\gamma^L\left\|\nabla_\theta\mathrm{TF}_L\left(Z_0;\theta_L^{\mathrm{TD}}\right)\right\|_1\right]\qquad\text{(Lemma 4)}$$

$$\leq\mathbb{E}_{s_q\sim\mu^p,p\sim\Delta(p),r\sim\Delta(r)}\left[\|v\|_\infty\gamma^L\left(\frac{L^2+L}{2}\nu+L\xi\right)\right]\qquad\text{(Lemma 3)}$$

$$=\mathbb{E}_{p\sim\Delta(p),r\sim\Delta(r)}[\|v\|_\infty]\gamma^L\left(\frac{L^2+L}{2}\nu+L\xi\right).$$

Let $C_v\doteq\mathbb{E}_{p\sim\Delta(p),r\sim\Delta(r)}[\|v\|_\infty]$, we have $J'(\theta_L^{\mathrm{TD}})\leq\frac{L^2+L}{2}\nu C_v\gamma^L+L\xi C_v\gamma^L$. Therefore, it holds that

$$\lim_{L\to\infty}J'(\theta_L^{\mathrm{TD}})\leq\lim_{L\to\infty}\frac{L^2+L}{2}\nu C_v\gamma^L+L\xi C_v\gamma^L=0$$

because the exponential term $\gamma^L$ dominates the polynomial terms of $L$. Since $J'(\theta_L^{\mathrm{TD}})$ is a norm and thus always nonnegative, we have

$$\lim_{L\to\infty}J'(\theta_L^{\mathrm{TD}})=0,$$

completing the proof. □

## C  Experiment Details

### C.1  Boyan's Chain

We include an illustration of Boyan's chain and its generation method here.

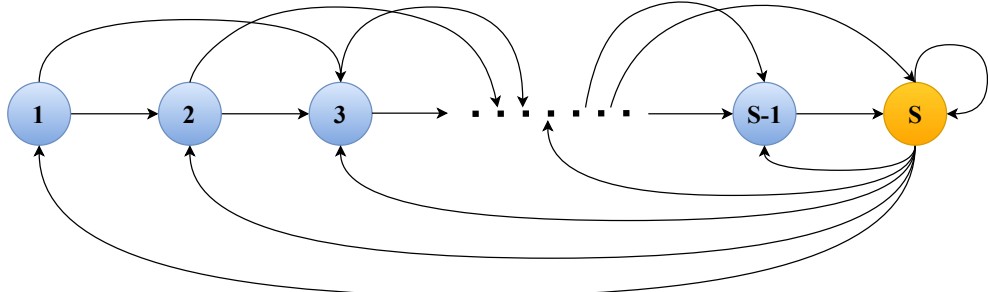

Figure 3: Boyan's chain with $S$ states. Arrows indicate non-zero transition probabilities.

### C.2  Compute Resources

We run our experiments in parallel on a single node of a CPU cluster. The node has 150 CPU cores and 150 GB of memory. The wall clock time it takes to finish running the experiments is about 50 minutes for multi-task TD and 15 hours for multi-task MC.

### C.3  Hyperparameters and More Implementation Details

We use NumPy [Harris et al., 2020] for data processing and implementing the MRPs. We use PyTorch [Ansel et al., 2024] to create and train our models. For data visualization, we use Matplotlib [Hunter, 2007] to create the plots. Table 1 lists the hyperparameters we used for experimentation.

**Algorithm 3:** Boyan Chain MRP Generation (Adapted from Algorithm 2 of Wang et al. [2025])

---

1: **Input:** state space size $n = |\mathcal{S}|$
2: $r \sim \text{Uniform}\left[(-1,1)^n\right]$  `// reward function`
3: $p \leftarrow 0_{m \times m}$  `// transition function`
4: **for** $i = 1, \ldots, n-2$ **do**
5:   $\epsilon \sim \text{Uniform}\left[(0,1)\right]$
6:   $p(i, i+1) \leftarrow \epsilon$
7:   $p(i, i+2) \leftarrow 1 - \epsilon$
8: **end for**
9: $p(n-1, n) \leftarrow 1$
10: $z \sim \text{Uniform}\left[(0,1)^n\right]$
11: $z \leftarrow z / \sum_s z(s)$
12: $p(n, 1:n) \leftarrow z$
13: $p_0 \leftarrow \texttt{stationary\_distribution}(p)$  `//initial distribution`
14: **Output:** MRP $(p_0, p, r)$

---

| optimizer | Adam [Kingma and Ba, 2015] |
|---|---|
| learning rate | 0.001 |
| weight decay | 0.0 |
| batch size | 64 |
| # of attention layers | 30 |
| # of Boyan's chain states | 5 |
| discount factor | 0.9 |
| # of Monte Carlo rollout steps | 200 |
| # of random seeds | 20 |
| # of Boyan's chain tasks for training | 20,000 |
| # of validation instances | 10 |
| validation context lengths | 5, 10, ..., 100 |

Table 1: Hyperparameters and more training details.

# D  Hyperparameter Study

## D.1  Transformer Depth

We conducted additional experiments to investigate the robustness of the emergence of ICTD under different numbers of attention layers. Keeping all other configurations unchanged, we pretrained a 5-layer, a 10-layer, and a 60-layer Transformer for 20,000 tasks. Each experiment was repeated for 20 times. Figure 4 plots the mean converged weights of the Transformers. We observe that the pattern of $\theta^{\text{TD}}$ emerged in all the tested depths for both multi-task TD and MC. We further tested the converged Transformers on unseen MRPs and report the mean and standard error of the MSVE as a function of the context length, as displayed in Figure 5. We witness a steady decrease of the MSVE as the context length grows, demonstrating clear in-context RL capability of the pretrained models.

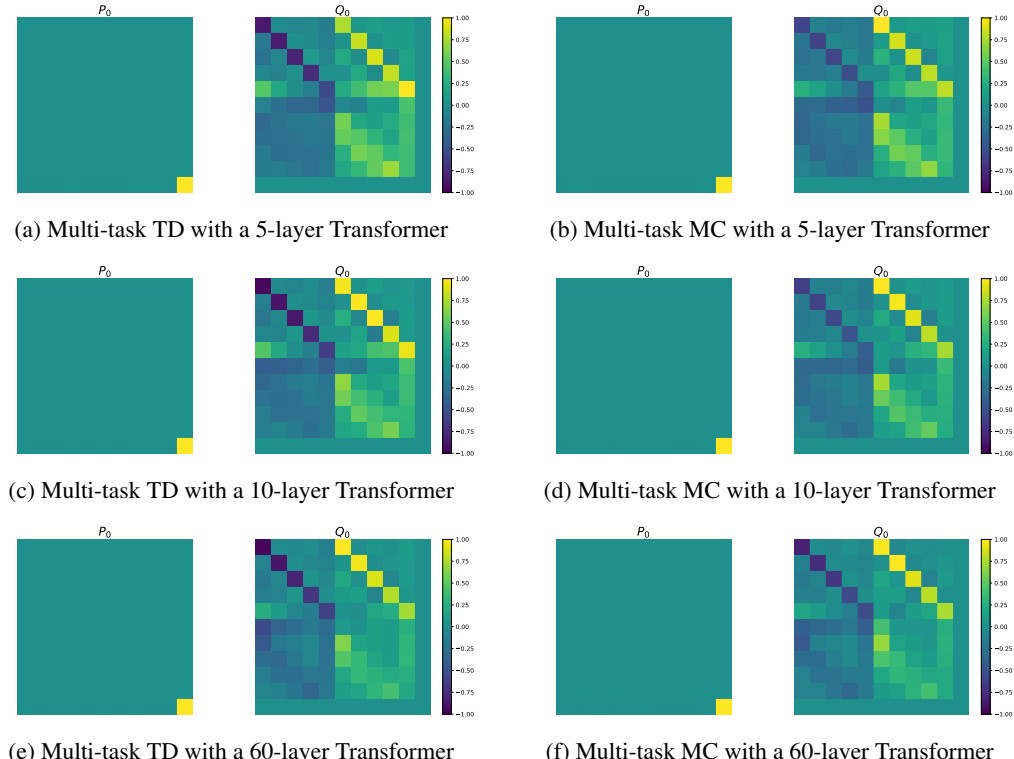

(a) Multi-task TD with a 5-layer Transformer

(b) Multi-task MC with a 5-layer Transformer

(c) Multi-task TD with a 10-layer Transformer

(d) Multi-task MC with a 10-layer Transformer

(e) Multi-task TD with a 60-layer Transformer

(f) Multi-task MC with a 60-layer Transformer

Figure 4: Mean parameters of 5-, 10-, and 60-layer Transformers after pretraining. Parameters are averaged over 20 trials and normalized to lie within the range $[-1, 1]$

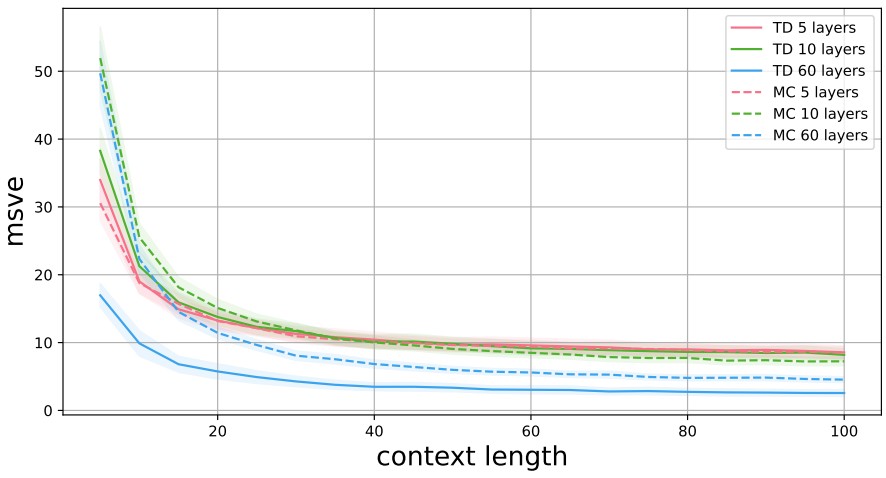

Figure 5: Mean and standard error of the averaged MSVEs against context lengths for 5-, 10-, and 60-layer Transformers. The curves are averaged over 20 random trials. The shaded areas represent the standard errors.

## D.2 Number of States

We also studied the emergence of ICTD under different numbers of states of the MRPs for training. Fixing all other configurations, we additionally pretrained the Transformer on Boyan's chains with 3 and 10 states, respectively. Notably, since we employ state enumeration for **training**, the context length is consistent with the number of states of the MRP during **training**. Hence, this experiment also ablates different training context sizes. Each experiment was repeated for 20 independent trials. Figure 6 plots the mean converged weights of the Transformers. All of them have a clear pattern of $\theta^{\text{TD}}$. Figure 7 illustrates the decreasing MSVEs vs. the increasing context length for the converged models in the test set. Again, all the pretrained models exhibit in-context policy evaluation behaviors.

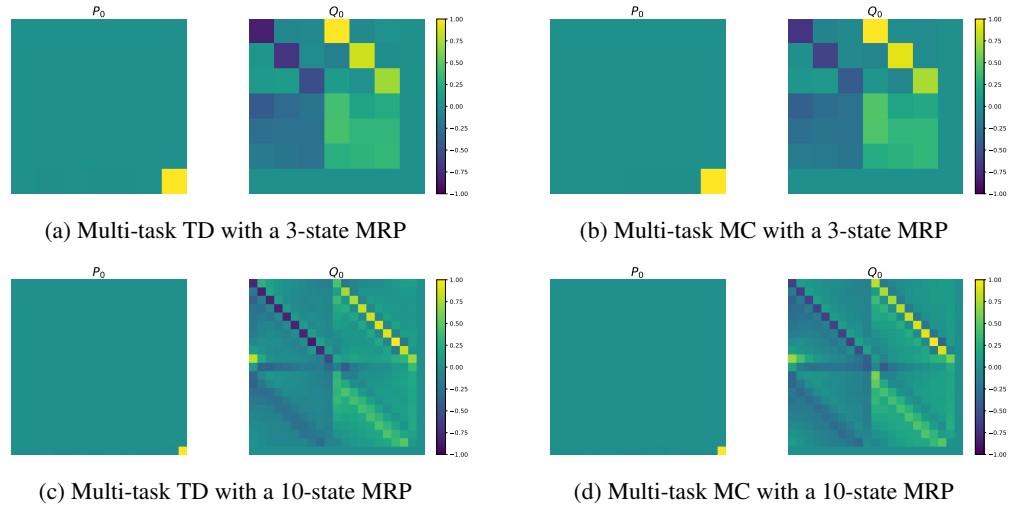

(a) Multi-task TD with a 3-state MRP    (b) Multi-task MC with a 3-state MRP

(c) Multi-task TD with a 10-state MRP    (d) Multi-task MC with a 10-state MRP

Figure 6: Mean parameters of the Transformers after pretraining on 3- and 10-state MRPs. Parameters are averaged over 20 trials and normalized to lie within the range $[-1, 1]$

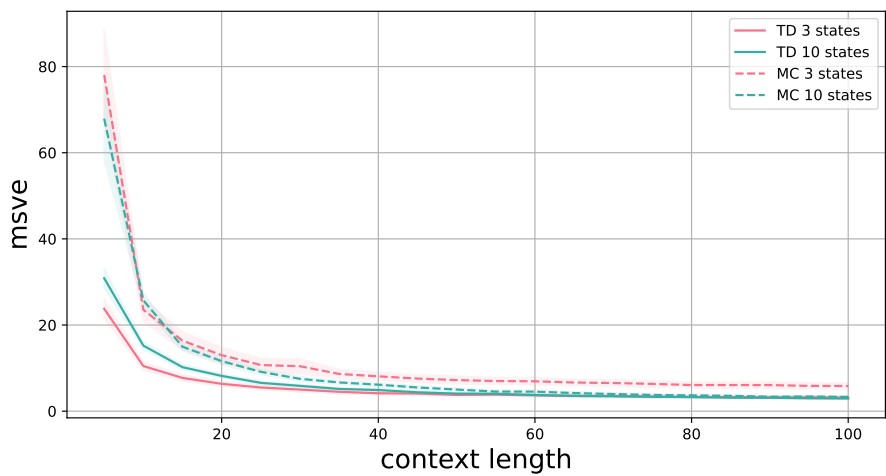

Figure 7: Mean and standard error of the averaged MSVEs against context lengths for Boyan's chains with 3 and 10 states, respectively. The curves are averaged over 20 random trials. The shaded areas represent the standard errors.

### D.3 MRP

Since we hypothesize that training with multi-task TD and MC induces ICTD capabilities in the Transformers, and since TD is not environment-specific, then the Transformers pretrained on one class of MRPs should be capable of predicting the values of another class of MRPs. In our last ablation study, we pretrained the Transformers on the Loop environment used by Wang et al. [2025] and evaluated them on the data generated by the Boyan's chain MRPs. Algorithm 4 sketches the generation procedure of the Loop MRPs, and Figure 8 is an illustrative example of a 6-state Loop environment. Figure 9 shows the mean converged parameters of the Transformers, supporting the emergence of $\theta^{\text{TD}}$ for Transformers pretrained on the Loop MRPs. In addition, Figure 10 plots the consistent decrease of the MSVEs against the context length on the tested Boyan's chain environments, supporting our hypothesis that the Transformer is not overfitting to one particular class of MRPs but learning a general prediction algorithm in its forward pass via pretraining with multi-task TD or MC.

---

**Algorithm 4:** Loop MRP Generation

1: **Input:** state space size $n = |\mathcal{S}|$; threshold $\epsilon \in (0, 1)$
2: $r \sim \text{Uniform}\left[(-1, 1)^n\right]$  // reward function
3: $c \sim \text{Uniform}\left[(0, 1)^{n \times n}\right]$  // connectivity
4: $c \leftarrow \mathbb{I}\{c > \epsilon\}$  // apply indicator function element-wise
5: **for** $i = 1, \ldots n - 1$ **do**
6:   $c(i, i) \leftarrow 0$  // no self-loop
7:   $c(i, i + 1) \leftarrow 1$  // can transition to next state
8: **end for**
9: $c(n, n) \leftarrow 0$
10: $c(n, 1) \leftarrow 1$
11: $p \sim \text{Uniform}\left[(0, 1)^{n \times n}\right]$  // transition function
12: $p \leftarrow p * c$  // element-wise multiplication
13: **for** $i = 1, \ldots, n$ **do**
14:   $z \leftarrow \sum_{j=1}^{n} p(i, j)$  // normalizing constant
15:   $p(i, 1 : n) \leftarrow p(i, 1 : n)/z$
16: **end for**
17: $p_0 \leftarrow \texttt{stationary\_distribution}(p)$  //initial distribution
18: **Output:** MRP $(p_0, p, r)$

---

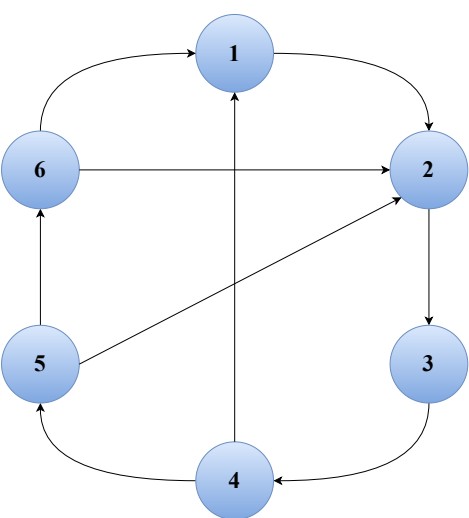

Figure 8: Loop with 6 states. Arrows indicate non-zero transition probabilities.

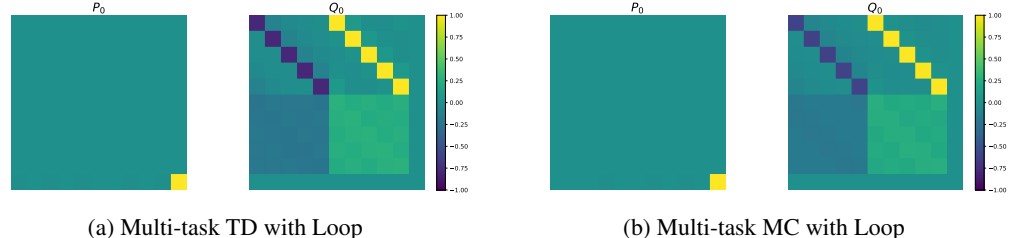

(a) Multi-task TD with Loop             (b) Multi-task MC with Loop

Figure 9: Mean parameters of the Transformers after pretraining on the Loop MRPs. Parameters are averaged over 20 trials and normalized to lie within the range $[-1, 1]$

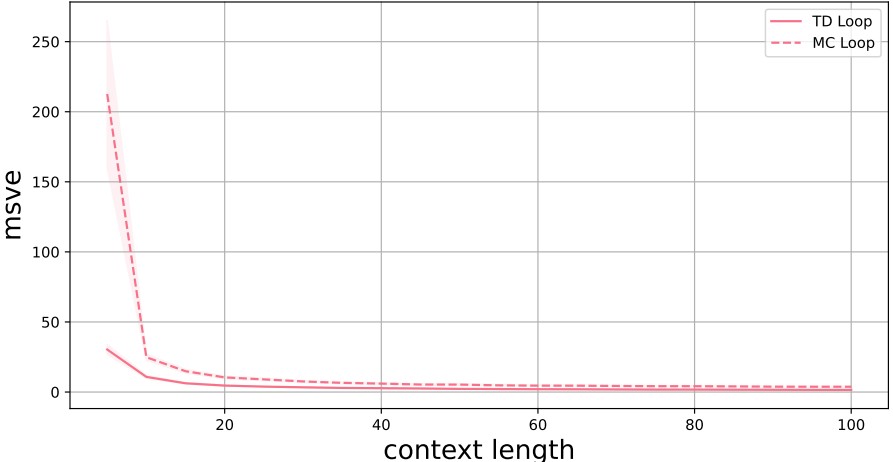

Figure 10: Mean and standard error of the averaged MSVEs against context lengths for Boyan's chains trained on the Loop MRPs. The curves are averaged over 20 random trials. The shaded areas represent the standard errors.

