# OpenReview forum: "Towards Provable Emergence of In-Context Reinforcement Learning"
_NeurIPS.cc/2025/Conference — NeurIPS 2025 poster_

### Official Review · Reviewer_xY5j · 2025-06-24

**Clarity:** 2
**Significance:** 2
**Originality:** 3
**Rating:** 4
**Confidence:** 3

**Summary:**

This paper investigates in-context reinforcement learning using Transformers with linear attention. Specifically, the authors establish two theoretical results: (1) In finite state spaces and under a particular sparse structure of the weight matrix, a specially parameterized prompt—equipped with known transition and reward functions—can compute the value function when passed through the Transformer (Theorem 1). (2) The desired weight matrix is shown to be a global minimizer of a pretraining loss function, where gradient descent is performed and the transition and reward components in the prompt are sampled from a prior distribution (Theorem 2). The paper also includes a synthetic experiment that numerically validates these two theorems.

**Questions:**

- I understand that by "multi-tasking," the paper refers to sampling different ground-truth transition and reward functions and then computing the value function under each sampled environment. Is this interpretation correct? If so, does this find application in other reinforcement learning studies?
- In Section 6.1, it appears that the trained Transformer is able to perform policy evaluation accurately even when provided with only a single realization (i.e., a sample trajectory) from the task environment. This seems to differ from the setting described in Equation (4), where the prompt consists of an enumeration of the full state space. I am curious whether this discrepancy has significant empirical or theoretical implications. Specifically, I would expect that unless the stationary distribution is uniform over the state space, such an approximation could introduce bias. Could the authors comment on this distinction?

**Ethical Concerns:**

["NO or VERY MINOR ethics concerns only"]

**Final Justification:**

My concerns about the paper's clarity and the structural assumptions on the transformer are addressed in the rebuttal. The concern about whether the setup is overly toy-like and misaligned with real-world RL or ICRL applications is partially addressed—while related literature exists, I am not aware of ICRL application papers that adopt this exact setting. Overall, I lean toward accepting the paper, though it remains a borderline case.

**Limitations:**

See weaknesses above.

**Paper Formatting Concerns:**

No such concerns.

**Quality:**

3

**Strengths And Weaknesses:**

Strengths

- The paper is generally well-structured and easy to follow.
- The theoretical investigation of in-context reinforcement learning is novel and addresses an emerging problem of interest.

Weaknesses

- The paper assumes a finite state space and full knowledge of the ground-truth transition and reward functions. Under these assumptions, the reinforcement learning task reduces to computing the value function of an infinite-horizon Markov reward process, which is arguably not representative of core challenges in modern RL and can be solved by relatively simple methods.
- The writing contains some inconsistencies and relies on setups or results from prior work without sufficient explanation. For example, in line 197, the prompt appears to be derived from a trajectory of length $n$, whereas elsewhere $n$ denotes the size of the state space. Additionally, the "multi-task TD" setting is introduced and used in experiments without a clear formal definition.
- As acknowledged in the limitations section, the Transformer architecture and the sparse parameterization considered in this work represent a highly simplified setting that lacks scalability to realistic applications.

---

> ### Author Rebuttal · Authors · 2025-07-30
>
> We sincerely thank the reviewer for the constructive feedback and detailed questions. We thank the reviewer for acknowledging the novelty of our theoretical investigation and the quality of our writing. We hope our response below can address the concerns and the questions.
>
> > Under these assumptions, the reinforcement learning task reduces to computing the value function of an infinite-horizon Markov reward process, which is arguably not representative of core challenges in modern RL
>
> We agree that we worked with a simplified RL setting. Nevertheless, we still preserve the challenge of temporal credit assignment (i.e., backward propagation of the rewards) -- a core challenge of modern RL that distinguishes our work from in-context regression. So we argue that our results still constitute a big leap over SOTA results on understanding in-context learning and in-context RL. The MRP formulation helps us to isolate the challenge of temporal credit assignment. Empirically, it allows us to easily compute the ground truth value function and the stationary distribution, which enables us to compute the MSVE accurately. We would not have these benefits if we were to work with control.
>
> > and can be solved by relatively simple methods.
>
> We agree that our simplified problem setting can be solved by other relatively simple methods more efficiently. However, our goal is not to create a new SOTA for solving this simplified problem. Instead, we aim to advance the theoretical understanding of the in-context TD phenomenon first established by [2] and provide a finer explanation for why reinforcement pretraining, like multi-task TD and MC, can induce in-context policy evaluation capabilities in Transformers from a training dynamics point of view. We argue that this simplified model does serve this purpose well and creates new insights.
>
> > The writing contains some inconsistencies and relies on setups or results from prior work without sufficient explanation.
>
> We apologize for the confusion caused by our notation inconsistency and insufficient explanations. We will update our manuscript to refrain from reusing the same notations for different meanings across sections. In our setup, $n = |\mathcal{S}|$ because the context consists of an enumeration of the states. However, we agree that we shouldn't have used $n$ to also denote the sampled trajectory length earlier. We will also add more explanations of the setup and an explicit definition of multi-task TD inherited from [2].
>
> > the Transformer architecture and the sparse parameterization considered in this work represent a highly simplified setting that lacks scalability to realistic applications.
>
> We acknowledge that our assumptions about the Transformer leave a gap between our theory and practice. In our defence, we would like to note that such assumptions are common and standard in works that aim to whitebox the in-context learning algorithms and explain their emergence in Transformers. For instance, notable works like [3], [4], [5], [6], [7], and [8] all assumed linear attention in their analysis. Furthermore, [5], [7], and [8] enforced the sparse parameter constraint. Without such assumptions, the manual computation of the forward propagation and the gradients of a multi-layer Transformer becomes intractably convoluted. In regard to the scalability to realistic applications, we argue that our goal is not to propose a state-of-the-art RL algorithm but to advance the explanability of the emergence of in-context RL as observed in large Transformer-based models [9].
>
> > I understand that by "multi-tasking," the paper refers to sampling different ground-truth transition and reward functions and then computing the value function under each sampled environment. Is this interpretation correct?
>
> The reviewer's interpretation of multi-tasking is correct. By multi-tasking, we meant training the Transformer on multiple MRPs, each having its unique transition dynamics and reward function.
>
> > If so, does this find application in other reinforcement learning studies?
>
> Multi-task training is an important technique in meta-RL, where the agent trained on multiple tasks achieves generalization and exhibits zero-shot/few-shot learning capabilities. A notable early work is RL$^2$ [10]. See [11] for a more comprehensive survey of how such multi-task training is useful in meta RL. One can consider in-context RL algorithms to be a subset of meta RL algorithms. A distinctive feature of ICRL algorithms is that the agent achieves generalization via encoding an RL algorithm in the forward pass of its model.
>
> > In Section 6.1, it appears that the trained Transformer is able to perform policy evaluation accurately even when provided with only a single realization (i.e., a sample trajectory) from the task environment. This seems to differ from the setting described in Equation (4), where the prompt consists of an enumeration of the full state space.
>
> We thank the reviewer for carefully examining our manuscript and raising this question. We planned to examine the in-context policy evaluation capabilities of the Transformer by showing a decreasing MSVE against an increasing context length, analogous to Figure 1 of [2]. However, in our theoretical analysis, the context length has to match the number of states due to enumeration. Hence, if we were to choose to mirror our theoretical setup for evaluation, we would have to generate a different MRP for each context length. For instance, the MSVE reported on context length 5 would come from a different MRP than the one used to generate the context of length 10. In this case, it would not be an apple-to-apple comparison between the two MSVEs since they are computed for two different MRPs, thus defeating the purpose. Consequently, we must give up satisfying the state enumeration assumption and sample the contexts of different lengths from the same MRP for each validation trial. Recall that we proved $\theta^{TD}$ is a global minimum of the NEU loss for both multi-task TD and MC as the layer goes infinitely deep.  If the weights of the Transformer indeed converge to $\theta^{TD}$, then the model learns to implement TD learning in its forward pass. Since Wang et al. [2] generated their Figure 1 by unrolling the MRP, we choose to approximate their setting by directly sampling from the stationary distribution, as the state distribution would converge to the stationary distribution beginning from any distribution as the context gets longer.
>
> >Specifically, I would expect that unless the stationary distribution is uniform over the state space, such an approximation could introduce bias.
>
> Since we are using one-hot features, the value functions are fully representable by the features, which is equivalent to the tabular setting. Hence, as long as the context covers all the states, there would not be any bias in value estimation. So, as long as the sampling distribution supports the state space, there would not be any bias when the context is sufficiently long. As our MRPs are ergodic, the stationary distribution indeed supports the state space, which should be feasible for our evaluation purpose.
>
> [1] Sutton, R., Barto, A., & others (1998). Reinforcement learning: An introduction. (Vol. 1) MIT press Cambridge.
>
> [2] Jiuqi Wang, Ethan Blaser, Hadi Daneshmand, & Shangtong Zhang (2025). Transformers Can Learn Temporal Difference Methods for In-Context Reinforcement Learning. In The Thirteenth International Conference on Learning Representations.
>
> [3] Sander, M., Giryes, R., Suzuki, T., Blondel, M., & Peyré, G. (2024). How do transformers perform in-context autoregressive learning?. In Proceedings of the 41st International Conference on Machine Learning.
>
> [4] Chenyu Zheng, Wei Huang, Rongzhen Wang, Guoqiang Wu, Jun Zhu, & Chongxuan Li (2024). On Mesa-Optimization in Autoregressively Trained Transformers: Emergence and Capability. In The Thirty-eighth Annual Conference on Neural Information Processing Systems.
>
> [5] Kwangjun Ahn, Xiang Cheng, Hadi Daneshmand, & Suvrit Sra (2023). Transformers learn to implement preconditioned gradient descent for in-context learning. In Thirty-seventh Conference on Neural Information Processing Systems.
>
> [6] Von Oswald, J., Niklasson, E., Randazzo, E., Sacramento, J., Mordvintsev, A., Zhmoginov, A., & Vladymyrov, M. (2023). Transformers learn in-context by gradient descent. In Proceedings of the 40th International Conference on Machine Learning.
>
> [7] Khashayar Gatmiry, Nikunj Saunshi, Sashank J. Reddi, Stefanie Jegelka, & Sanjiv Kumar (2024). Can Looped Transformers Learn to Implement Multi-step Gradient Descent for In-context Learning?. In Forty-first International Conference on Machine Learning.
>
> [8] Zhang, R., Frei, S., & Bartlett, P. (2024). Trained transformers learn linear models in-context. J. Mach. Learn. Res., 25(1).
>
> [9] Chanwoo Park, Xiangyu Liu, Asuman E. Ozdaglar, & Kaiqing Zhang (2025). Do LLM Agents Have Regret? A Case Study in Online Learning and Games. In The Thirteenth International Conference on Learning Representations.
>
> [10] Yan Duan, John Schulman, Xi Chen, Peter L. Bartlett, Ilya Sutskever, & Pieter Abbeel. (2017). RL$^2$: Fast Reinforcement Learning via Slow Reinforcement Learning.
>
> [11] Jacob Beck, Risto Vuorio, Evan Zheran Liu, Zheng Xiong, Luisa Zintgraf, Chelsea Finn, & Shimon Whiteson. (2025). A Tutorial on Meta-Reinforcement Learning.

---

> > ### Comment · Reviewer_xY5j · 2025-08-04
> >
> > I thank the authors for their detailed rebuttal. However, my concern still remains with the setup of this work. Specifically, the assumption of access to the ground-truth reward and transition functions, as well as a fixed policy, seems misaligned with practical reinforcement learning applications. I do not feel this concern has been fully addressed in the rebuttal.
> >
> > To further persuade me to lean toward accepting this work, I would appreciate clarification on the following two points:
> >
> > 1.  Could the authors point to prior literature on in-context reinforcement learning applications (preferably with large-scale experiments) that adopt a similar setting to this work? While I understand that theoretical works often rely on simplified assumptions, it would be helpful to see precedents where transition and reward functions are used as part of the prompt for policy evaluation.
> >
> > 2. Can the proposed theory generalize to the setting of MDPs with non-fixed policies? For instance, at inference time, can the transformer—when prompted with environment information—evaluate the value function of *any* policy?
> >
> > Additionally, I did not fully follow the authors’ response regarding Section 6.1. In the equation between lines 321–322, the states $( S_0^{(i)}, \ldots, S_m^{(i)} )$ are randomly sampled. Therefore, it is possible that this set is not an enumeration of all states of task $i$. Then why is there no bias?

---

> > > ### Author Response · Authors · 2025-08-05
> > > **Reply to Official Comment Part I**
> > >
> > > We appreciate the reviewer for the stimulating follow-up questions, which help us refine the paper further. We provide more clarification below. We will integrate all the clarifications in the revision.
> > >
> > > >Specifically, the assumption of access to the ground-truth reward and transition functions, as well as a fixed policy, seems misaligned with practical reinforcement learning applications.
> > > >
> > > > Could the authors point to prior literature on in-context reinforcement learning applications (preferably with large-scale experiments) that adopt a similar setting to this work? ...
> > >
> > > We agree with the reviewer that assuming access to true transition and reward functions during test time is misaligned with practitioners. However, we want to note that this is entirely optional, and we used this setting mostly to align with the pretraining setup. Below, we present some new results (revision of Figure 1 in the main context) in more practical settings. In particular, we don't change the pretraining, but in the test time, instead of using the true transition function to compute the expectation of the feature of the successor state, we use an unrolled trajectory to construct the context:
> > > Recall that the original context (lines 321-322) has the form
> > > $$
> > >  [ \phi(S_0), \cdots, \phi(S_{m-1})
> > > $$
> > > $$
> > >  \gamma \sum_{s'\in\mathcal{S}} p(s' \mid S_0)\phi(s'), \cdots, \gamma \sum_{s'\in\mathcal{S}} p(s' \mid S_{m-1})\phi(s')
> > > $$
> > > $$
> > >  R_1, \cdots, R_m ],
> > > $$
> > > where $S_i \sim \mu$ for $i = 0, 1, \dots, m-1$ and $R_{i+1} = r(S_i)$.
> > > We now unroll the sampled MRP to generate a trajectory $S_0, R_1, S_1, \dots, S_{m-1}, R_m, S_m$ and construct the context as
> > > $$
> > >     [\phi\left(S_0\right) , \phi\left(S_1\right) , \cdots , \phi\left(S_{m-1}\right)
> > > $$
> > >
> > > $$
> > >     \gamma \phi(S_1) , \gamma \phi(S_2) , \cdots , \gamma \phi(S_m)
> > > $$
> > > $$
> > >     R_1 , R_2 , \cdots , R_m],
> > > $$
> > > where $S_0 \sim \mu$, $S_{i+1}\sim p(\cdot \mid S_i)$, and $R_{i+1} = r(S_i)$. This form of context construction also aligns with the one used in [6]. Therefore, we do not use the true transition function in test time to report the MSVEs now. To summarize, the current setting in this response is
> > > * Pretraining: requiring access to the underlying transition function and reward function
> > > * Deployment/testing: standard RL setting, no access to true transition and reward functions; just sampled states and rewards
> > >
> > > There are lots of classical RL works that adopt a similar setting. Particularly, there is an active research area called **differentiable simulation**, e.g., [1-4]. The key idea is that during the training, they assume access to the true transition (reward) function and use the true transition (reward) function to compute policy gradients better. Most of them are targeted at robot applications, so we would consider them large-scale. DeepMind even has a dedicated (and popular) open-source repo that turns MuJoCo into a fully differentiable one [5]. Admittedly, how we use the true transition function is different from the differentiable simulator community. To our defence, we would argue that assuming access to the true transition (reward) function during training is somewhat an accepted norm by practitioners, as long as we don't make this assumption during evaluation. In-context RL is a relatively new area, so we are not aware of prior ICRL works that adopt a similar setting. Nevertheless, we believe the ICRL community will eventually catch up with the classical RL community to consider differentiable simulators and even more creative ways to leverage the transition function in the training, like this paper does.
> > >
> > > [1] Newbury, R., Collins, J., He, K., Pan, J., Posner, I., Howard, D., & Cosgun, A. (2024). A Review of Differentiable Simulators. IEEE Access, 12, 97581-97604.
> > >
> > > [2] Mora, M., Peychev, M., Ha, S., Vechev, M., & Coros, S. (2021). PODS: Policy Optimization via Differentiable Simulation. In Proceedings of the 38th International Conference on Machine Learning (pp. 7805–7817). PMLR.
> > >
> > > [3] Suh, H., Simchowitz, M., Zhang, K., & Tedrake, R. (2022). Do Differentiable Simulators Give Better Policy Gradients?. In Proceedings of the 39th International Conference on Machine Learning (pp. 20668–20696). PMLR.
> > >
> > > [4] Wiedemann, N., Wüest, V., Loquercio, A., Müller, M., Floreano, D., & Scaramuzza, D. (2023). Training Efficient Controllers via Analytic Policy Gradient. In 2023 IEEE International Conference on Robotics and Automation (ICRA) (pp. 1349-1356).
> > >
> > > [5] Freeman, C. D., Frey, E., Raichuk, A., Girgin, S., Mordatch, I., & Bachem, O. (2021). Brax - A Differentiable Physics Engine for Large Scale Rigid Body Simulation (Version 0.12.4).
> > >
> > > [6] Jiuqi Wang, Ethan Blaser, Hadi Daneshmand, & Shangtong Zhang (2025). Transformers Can Learn Temporal Difference Methods for In-Context Reinforcement Learning. In The Thirteenth International Conference on Learning Representations.

---

> > > ### Author Response · Authors · 2025-08-05
> > > **Reply to Official Comment Part II**
> > >
> > > We now present the updated Figure 1 under this new setting in a table. As shown in the table below, the policy evaluation error still decreases considerably as the context length grows. We note that our theoretical analysis about the global optimizer is an argument purely for the pretraining. Since we do not alter the pretraining stage, this new result is still closely related to our theory. Furthermore, [6] also has some results showing that this Transformer can perform policy evaluation on CartPole for any policies during inference time, without access to the underlying transition (reward) functions. We did not reproduce it to avoid repetition.
> > >
> > > | Context Length | TD | MC |
> > > |----------------|----|----|
> > > | 5 | 27.84 | 71.04 |
> > > | 10 | 15.26 | 27.45 |
> > > | 15 | 11.19 | 18.71 |
> > > | 20 | 8.94 | 14.62 |
> > > | 25 | 7.86 | 12.61 |
> > > | 30 | 7.22 | 11.17 |
> > > | 35 | 6.56 | 9.99 |
> > > | 40 | 6.09 | 9.38 |
> > > | 45 | 5.89 | 8.47 |
> > > | 50 | 5.63 | 8.11 |
> > > | 55 | 5.39 | 7.71 |
> > > | 60 | 5.18 | 7.04 |
> > > | 65 | 5.16 | 7.29 |
> > > | 70 | 5.03 | 6.91 |
> > > | 75 | 4.86 | 7.03 |
> > > | 80 | 4.78 | 6.65 |
> > > | 85 | 4.73 | 6.58 |
> > > | 90 | 4.61 | 6.46 |
> > > | 95 | 4.49 | 6.46 |
> > > | 100 | 4.44 | 6.18 |
> > >
> > > [1] Newbury, R., Collins, J., He, K., Pan, J., Posner, I., Howard, D., & Cosgun, A. (2024). A Review of Differentiable Simulators. IEEE Access, 12, 97581-97604.
> > >
> > > [2] Mora, M., Peychev, M., Ha, S., Vechev, M., & Coros, S. (2021). PODS: Policy Optimization via Differentiable Simulation. In Proceedings of the 38th International Conference on Machine Learning (pp. 7805–7817). PMLR.
> > >
> > > [3] Suh, H., Simchowitz, M., Zhang, K., & Tedrake, R. (2022). Do Differentiable Simulators Give Better Policy Gradients?. In Proceedings of the 39th International Conference on Machine Learning (pp. 20668–20696). PMLR.
> > >
> > > [4] Wiedemann, N., Wüest, V., Loquercio, A., Müller, M., Floreano, D., & Scaramuzza, D. (2023). Training Efficient Controllers via Analytic Policy Gradient. In 2023 IEEE International Conference on Robotics and Automation (ICRA) (pp. 1349-1356).
> > >
> > > [5] Freeman, C. D., Frey, E., Raichuk, A., Girgin, S., Mordatch, I., & Bachem, O. (2021). Brax - A Differentiable Physics Engine for Large Scale Rigid Body Simulation (Version 0.12.4).
> > >
> > > [6] Jiuqi Wang, Ethan Blaser, Hadi Daneshmand, & Shangtong Zhang (2025). Transformers Can Learn Temporal Difference Methods for In-Context Reinforcement Learning. In The Thirteenth International Conference on Learning Representations.

---

> > > ### Author Response · Authors · 2025-08-05
> > > **Reply to Official Comment Part III**
> > >
> > > >Can the proposed theory generalize to the setting of MDPs with non-fixed policies? For instance, at inference time, can the transformer—when prompted with environment information—evaluate the value function of any policy?
> > >
> > > Yes. In the new test setup above, the environment information we prompted the Transfomer with was simply the standard trajectory $S_0, R_1, S_1, R_2, S_2, \dots$. Note that [6] proved that the forward pass of this Transformer during inference is precisely mathematically equivalent to TD. Thus, during inference, we only need the same information as the TD algorithm, and the trained Transformer accomplishes the same as TD. Since TD can perform policy evaluation for any policy, this Transformer can also evaluate any policy during inference.
> > > Besides, this is also exactly what we demonstrate in our new result (which was also demonstrated in the old result). During evaluation, we randomly generated multiple MRPs, corresponding to multiple MDPs and policies (an MRP is just a pair of MDP and a policy), and tested the trained Transformer against all these MRPs. Notably, these test MRPs were never used for training. The steady decrease of the policy evaluation errors against the increasing context lengths in evaluation attests to the Transformer's capability to estimate the value function of not only any policy but also any transition functions, even if they are unseen during training, even if we do not provide the exact transition function and the exact policy to the Transformer.
> > >
> > > > Additionally, I did not fully follow the authors’ response regarding Section 6.1. In the equation between lines 321–322, the states are randomly sampled. Therefore, it is possible that this set is not an enumeration of all states of task. Then why is there no bias?
> > >
> > > We believe that the confusion stems from the difference in context construction between pretraining and testing. During pretraining, the context length is fixed to be the number of states of the MRP. In the test time, however, the context length grows longer and longer. The reviewer is correct in that for any fixed $m$, it's possible that the set does not enumerate all states. Whereas, as $m$ grows, it will almost surely eventually enumerate all states. The larger $m$, the more likely that it already enumerates all states. Therefore, we demonstrated that the policy evaluation error decreased with the increase of $m$ in our empirical study. We say it's unbiased in this sense. This situation is the same as in TD. Consider the tabular TD algorithm. We usually say it's unbiased. However, if we run it for only 1 step, it certainly won't give an accurate value estimation. To get an accurate value estimation, it needs 1) to go over all states at least once, and 2) to propagate the rewards backwards completely. Here, 1) is achieved by growing the context length $m$, and 2) is achieved by increasing the Transformer depth.
> > >
> > > [1] Newbury, R., Collins, J., He, K., Pan, J., Posner, I., Howard, D., & Cosgun, A. (2024). A Review of Differentiable Simulators. IEEE Access, 12, 97581-97604.
> > >
> > > [2] Mora, M., Peychev, M., Ha, S., Vechev, M., & Coros, S. (2021). PODS: Policy Optimization via Differentiable Simulation. In Proceedings of the 38th International Conference on Machine Learning (pp. 7805–7817). PMLR.
> > >
> > > [3] Suh, H., Simchowitz, M., Zhang, K., & Tedrake, R. (2022). Do Differentiable Simulators Give Better Policy Gradients?. In Proceedings of the 39th International Conference on Machine Learning (pp. 20668–20696). PMLR.
> > >
> > > [4] Wiedemann, N., Wüest, V., Loquercio, A., Müller, M., Floreano, D., & Scaramuzza, D. (2023). Training Efficient Controllers via Analytic Policy Gradient. In 2023 IEEE International Conference on Robotics and Automation (ICRA) (pp. 1349-1356).
> > >
> > > [5] Freeman, C. D., Frey, E., Raichuk, A., Girgin, S., Mordatch, I., & Bachem, O. (2021). Brax - A Differentiable Physics Engine for Large Scale Rigid Body Simulation (Version 0.12.4).
> > >
> > > [6] Jiuqi Wang, Ethan Blaser, Hadi Daneshmand, & Shangtong Zhang (2025). Transformers Can Learn Temporal Difference Methods for In-Context Reinforcement Learning. In The Thirteenth International Conference on Learning Representations.

---

> > > > ### Comment · Reviewer_xY5j · 2025-08-06
> > > >
> > > > Thank you for the detailed response. My concern about the fixed-policy assumption is partially addressed, though I suggest formally stating the corresponding result mathematically to improve clarity. The relevance of the setting to large-scale applications is also somewhat clarified by the connection to differentiable simulation. However, I recommend a more thorough review of the ICRL literature, particularly application-focused works, to better support this connection. As currently written, the cited ICRL papers do not appear to share the same setup.
> > > >
> > > > Given these clarifications, I am willing to raise my score. However, due to remaining uncertainty about how closely the setting aligns with practical ICRL applications, I will lower my confidence level.

---

> > > > > ### Author Response · Authors · 2025-08-06
> > > > >
> > > > > We thank the reviewer for their continued engagement, which helped us improve and refine our paper. We will clarify the distinction between our pretraining and evaluation settings and accompany our results with equations to enhance clarity. We will also cite more work in ICRL, particularly ones with applications, to better justify our setting.

---

### Official Review · Reviewer_N5DC · 2025-07-03

**Clarity:** 3
**Significance:** 2
**Originality:** 2
**Rating:** 4
**Confidence:** 3

**Summary:**

The paper tackles the question of why a Transformer that is pretrained with a standard reinforcement-learning algorithm can later exhibit in-context reinforcement learning (ICRL). Building directly on the single-layer analysis of Wang et al.​ (2025), it generalizes the setting to multi-layer linear-attention Transformers used for policy evaluation. The authors (1) prove an inference-time result: with a particular “ICTD’’ weight configuration the value-prediction error shrinks to zero as depth grows, so the forward pass implements batched temporal-difference (TD) learning; (2) establish that the same weights are global minimizers of a norm-of-expected-update loss for both multi-task TD and a newly introduced multi-task Monte-Carlo objective, offering an optimization-based explanation for their empirical emergence; and (3) confirm the theory on randomly generated tasks, where pretrained models improve steadily with longer context lengths and converge numerically to the predicted ICTD weights.

**Questions:**

How does MSVE change if you vary the number of layers? Can you elaborate on this with an empirical study?

**Ethical Concerns:**

["NO or VERY MINOR ethics concerns only"]

**Limitations:**

All theory and experiments rely on linear attention; real-world models use softmax non-linear operator.

**Paper Formatting Concerns:**

The paper is well organized and mathematically precise, but Sections 3-4 are notation-dense and could benefit from an intuitive walkthrough of the prompt construction.

**Quality:**

3

**Strengths And Weaknesses:**

The theoretical development is sound. Both the inference-time convergence proof and the global-minimizer result are stated with clear assumptions. The experiments, although lightweight, are cleanly targeted at the two main claims and reproduce the predicted behavior. A weakness is the heavy reliance on linear attention and one-hot features; practical relevance to nonlinear Transformers or continuous domains is still open.

---

> ### Author Rebuttal · Authors · 2025-07-30
>
> We greatly appreciate the reviewer for the honest review and thoughtful question. We are pleased to learn the compliments on the soundness of our theory and the effectiveness of our experiments. We hope our response below addresses the reviewer's concerns.
>
> >  A weakness is the heavy reliance on linear attention and one-hot features; practical relevance to nonlinear Transformers or continuous domains is still open.
>
> We acknowledge that linear attention is not widely adopted in modern Transformer architectures. Nevertheless, it is a standard simplifying assumption in establishing precise equivalence between the forward pass of the Transformer and some principled algorithm. Some notable theoretical works in in-context learning using linear attentions include [1], [2], [3], and [4]. Furthermore, as our work extends [5] to answer why ICTD emerges, we also need to follow their setup using linear attentions.
> As our work represents one of the earliest attempts to explain why ICTD emerges, to the best of our knowledge, we wish to begin with the tabular setting. As in many fundamental theoretical works in RL, the tabular setting lays out the foundation for studying linear function approximation by allowing us to isolate the core problem (i.e., the pretraining dynamics). The one-hot encoding is mathematically equivalent to the tabular representation.
>
> > How does MSVE change if you vary the number of layers?
>
> We conducted an additional hyperparameter study on Transformer depth in response to the reviewer's question. Keeping everything else unchanged, we additionally trained and evaluated 5-, 10-, and 60-layer Transformers for 20,000 tasks. We also repeated each experiment with 20 random seeds for statistical rigour. Let m denote the context length. Pasted below are the approximate mean MSVEs on the shortest context (m = 5) and the longest context (m = 100) tested across 5-, 10-, and 60-layer Transformers trained via multi-task TD and MC.
>
> Multi-task TD:
>
> |   | 5-layer | 10-layer | 60-layer |
> |---|---------|----------|----------|
> | m = 5| 12.0 | 7.0 | 2.1 |
> | m = 100 | 7.0 | 4.8 | 1.1|
>
> Multi-task MC:
> |   | 5-layer | 10-layer | 60-layer |
> |---|---------|----------|----------|
> | m = 5| 10.9| 7.8 | 3.0 |
> | m = 100 | 6.3 | 5.3 | 1.9
>
> We observe that in all cases, the mean MSVE dropped significantly when the context length increased, confirming the emergence of in-context policy evaluation. Furthermore, we notice that the deeper the Transformer, the better the prediction. Since under $\theta^{TD}$, one loop over the attention layer is equivalent to one step of TD update, we conjecture that the deeper Transformers apply more steps of TD updates internally, thus exhibiting better convergence. Lastly, we observe that the magnitudes of the mean MSVEs are of the same order between multi-task TD and MC. This observation provides further evidence that the Transformers trained via multi-task TD and MC are implementing the same in-context policy evaluation algorithm, at least behaviorally. It aligns with our theoretical result that $\theta^{TD}$ is a global minimizer of the NEU loss for both multi-task TD and MC. Please also consult our response to Reviewer De38 for more hyperparameter study outcomes.
>
> [1] Sander, M., Giryes, R., Suzuki, T., Blondel, M., & Peyré, G. (2024). How do transformers perform in-context autoregressive learning?. In Proceedings of the 41st International Conference on Machine Learning.
>
> [2] Chenyu Zheng, Wei Huang, Rongzhen Wang, Guoqiang Wu, Jun Zhu, & Chongxuan Li (2024). On Mesa-Optimization in Autoregressively Trained Transformers: Emergence and Capability. In The Thirty-eighth Annual Conference on Neural Information Processing Systems.
>
> [3] Kwangjun Ahn, Xiang Cheng, Hadi Daneshmand, & Suvrit Sra (2023). Transformers learn to implement preconditioned gradient descent for in-context learning. In Thirty-seventh Conference on Neural Information Processing Systems.
>
> [4] Von Oswald, J., Niklasson, E., Randazzo, E., Sacramento, J., Mordvintsev, A., Zhmoginov, A., & Vladymyrov, M. (2023). Transformers learn in-context by gradient descent. In Proceedings of the 40th International Conference on Machine Learning.
>
> [5] Jiuqi Wang, Ethan Blaser, Hadi Daneshmand, & Shangtong Zhang (2025). Transformers Can Learn Temporal Difference Methods for In-Context Reinforcement Learning. In The Thirteenth International Conference on Learning Representations.

---

> > ### Comment · Reviewer_N5DC · 2025-08-03
> >
> > thanks for addressing the questions!

---

> > > ### Author Response · Authors · 2025-08-05
> > >
> > > We thank the reviewer again for reviewing our work and raising insightful questions.

---

### Official Review · Reviewer_zvwM · 2025-07-05

**Clarity:** 3
**Significance:** 3
**Originality:** 3
**Rating:** 4
**Confidence:** 2

**Summary:**

This paper investigates the emergence of in-context reinforcement learning (ICRL) capabilities in Transformers pretrained with standard RL algorithms/objectives. The core question is: Why do RL-pretrained network parameters enable ICRL (i.e., solving new tasks without parameter updates by conditioning on context)? The authors hypothesize that such parameters are minimizers of the pretraining loss. Through a focused case study on policy evaluation, they prove the hypothesis.

**Questions:**

see above

**Ethical Concerns:**

["NO or VERY MINOR ethics concerns only"]

**Final Justification:**

After carefully reading the author's response, I have decided to keep my initial score.

**Limitations:**

yes

**Paper Formatting Concerns:**

no paper formatting concerns

**Quality:**

3

**Strengths And Weaknesses:**

Strengths:

1. To the best of mt knowledge, this paper offers the first proof that ICRL parameters (specifically for ICPE) emerge as global minimizers of standard RL pretraining losses (NEU). This directly addresses the "why" of emergent ICRL.

2. Authors provide a unified analysis framework that shows $\theta^{TD}_{L}$ minimizes NEU for both TD and MC pretraining (Theorem 2, Corollary 1), suggesting robustness across algorithms.

3. I appreciate that authors clearly define the scope and limitations of the proposed theory: they explicitly acknowledge assumptions (linear attention, enumerated states) and scalability limits (Sec 7).

Weaknesses:

1. Limited Practical Applicability: Theoretical results rely on linear Transformers, while practical ICRL uses nonlinear models (e.g., ReLU), leading to a significant gap in real-world applicability.

2. Small-Scale Experiments: Experiments use tiny MRPs (Boyan’s chain with 13 states). Claims lack validation in complex environments. And the experiment misses a comparison to non-ICRL baselines (e.g., fine-tuning/offline-to-online RL).

---

> ### Author Rebuttal · Authors · 2025-07-30
>
> We deeply thank the reviewer for reviewing our work and acknowledging the robustness of our core theoretical contribution. We hope the following can address the concerns of the reviewer.
>
> > Limited Practical Applicability: Theoretical results rely on linear Transformers, while practical ICRL uses nonlinear models (e.g., ReLU), leading to a significant gap in real-world applicability.
>
> We acknowledge that this assumption leaves a gap between the theory and the canonical form of the modern Transformer architecture. However, linear attention is one of the most common simplifying assumptions for establishing exact equivalence between the forward pass of the Transformer and some principled algorithm. Some notable works include [1], [2], [3], and [4]. Furthermore, as the majority of our work builds upon [5], where they prove linear Transformers can and do implement TD learning in-context, we very much need to inherit their setup because we need to work with in-context TD.
>
> > Experiments use tiny MRPs (Boyan’s chain with 13 states). Claims lack validation in complex environments.
>
> Since our primary goal is to empirically confirm our answer to why ICTD emerges, we need to mirror the setup of our theories. Thus, it simultaneously requires us to generate a large number of MRPs and have full access to their information. This requirement is not achievable with complex environments. On the other hand, having full control over the environment has several benefits. For instance, we can analytically solve for the ground truth value function and the stationary distribution of the MRP, allowing us to compute the MSVE against the ground truth. With a complex environment, one can only approximate it by unrolling the policy for many steps, which is neither efficient nor accurate. To compensate for the lack of complexity in our tasks, we ran additional hyperparameter studies to demonstrate that the emergence of ICTD is robust, as presented in the response to Reviewer De38.
>
> > And the experiment misses a comparison to non-ICRL baselines (e.g., fine-tuning/offline-to-online RL).
>
> We would like to claim that our goal in this work is not to propose a state-of-the-art RL algorithm but to provide an explanation for the emergence of in-context TD in Transformers through reinforcement pretraining, such as multi-task TD and MC. Thus, we design our experiments to verify our theory. Since other potential baselines are irrelevant to the goal of explanation, we do not include them in our empirical study.
>
> [1] Sander, M., Giryes, R., Suzuki, T., Blondel, M., & Peyré, G. (2024). How do transformers perform in-context autoregressive learning?. In Proceedings of the 41st International Conference on Machine Learning.
>
> [2] Chenyu Zheng, Wei Huang, Rongzhen Wang, Guoqiang Wu, Jun Zhu, & Chongxuan Li (2024). On Mesa-Optimization in Autoregressively Trained Transformers: Emergence and Capability. In The Thirty-eighth Annual Conference on Neural Information Processing Systems.
>
> [3] Kwangjun Ahn, Xiang Cheng, Hadi Daneshmand, & Suvrit Sra (2023). Transformers learn to implement preconditioned gradient descent for in-context learning. In Thirty-seventh Conference on Neural Information Processing Systems.
>
> [4] Von Oswald, J., Niklasson, E., Randazzo, E., Sacramento, J., Mordvintsev, A., Zhmoginov, A., & Vladymyrov, M. (2023). Transformers learn in-context by gradient descent. In Proceedings of the 40th International Conference on Machine Learning.
>
> [5] Jiuqi Wang, Ethan Blaser, Hadi Daneshmand, & Shangtong Zhang (2025). Transformers Can Learn Temporal Difference Methods for In-Context Reinforcement Learning. In The Thirteenth International Conference on Learning Representations.

---

> > ### Comment · Reviewer_zvwM · 2025-08-05
> >
> > Thank you for your response! I have no more questions. After reading the other reviews and responses, I will be keeping my score at 4.

---

> > > ### Author Response · Authors · 2025-08-05
> > >
> > > We are pleased to learn that we addressed the reviewer's concerns. We thank the reviewer for the feedback and the positive assessment of our work.

---

### Official Review · Reviewer_De38 · 2025-07-16

**Clarity:** 2
**Significance:** 3
**Originality:** 3
**Rating:** 4
**Confidence:** 4

**Summary:**

This is a theoretical study of the In-Context Policy Evaluation — a specific problem in the broader In-Context Reinforcement Learning setting. This setting allows for ground-up studying the emergence of ICRL capabilities in the forward passes of the transformer.

The authors develop a theory demonstrating the convergence of the TD and MC based approaches to be implemented in the forward pass of the linear transformer, as the number of layers goes to infinity (with empirical results covering depth 30 with looped weights).

Authors use Boyan’s chain to empirically verify their theoretical findings, demonstrating that convergence can happen in toy settings.

**Questions:**

First, I believe this paper is timely and important. However, at the moment, this paper falls for me into the borderline accept category. I do not see how it could be qualified for Strong Accept as I do not qualify it “with groundbreaking impact on one or more areas of AI”. But, with a certain amount of modifications, I would be willing to put the paper into the Accept category.

**What would make me to put the paper into the *Accept* category:**
- A proof of convergence. I understand that this is not an easy task, but that would solidify it as accept for me.
- An extended set of empirical investigation. Ablating the following would highly likely will result in moving your paper towards the accept:
  - ablation to different depths, e.g.: 4, 8, 16, 32, 64, etc
  - ablation for different context lengths: What happens for very small windows (m <= 5) or large ones (m>100)? Also is it possible to mismatch settings -- train with short context, evaluate with long?
  - experiments beyond Boyan Chains, not sure what this could be, but going beyond just one environment is a must

**What would make me to put the paper into the *Reject* category:**
- As the paper is primarily theoretical, the correctness of the proofs is a necessary and critical condition for acceptance. I have been able to only partially verify the proofs, and have not found any mistakes. However, if other reviewers would highlight any mistakes (that are difficult to fix within this review period), I would be inclined to reject the paper regardless of any experimental evaluations.

**Ethical Concerns:**

["NO or VERY MINOR ethics concerns only"]

**Final Justification:**

- The main reason to accept is the good theoretical development of the In-Context Reinforcement Learning.
- However, the limited scope of experiments which is induced by the theoretical limitations is rather difficult to ignore to assess the paper as having a major impact on at least one sub-area of AI, therefore borderline accept.

**Limitations:**

yes

**Paper Formatting Concerns:**

No major formatting concerns.

**Quality:**

3

**Strengths And Weaknesses:**

*Strengths*

- **Rigorous theoretical extension:** Builds on Wang et al. 2025 by generalizing the single-layer linear-attention result to deep (number of layers to infinity) linear transformers, with a formal bound showing value-prediction error decays exponentially in depth.
- **Unified global minimizer proof:** Introduces a novel analysis demonstrating that both Monte-Carlo and TD objectives share the same global minimizer under the NEU loss, tightening the link between the two training regimes.
- **Sharper characteristic of the solution set:**  Shows that the broad invariant set identified by Wang et al. 2025 collapses to a single point in the deep-network limit, eliminating extraneous parameter configurations.

*Weaknesses*
- **Remaining convergence gap:**  The key open question from Wang et al. whether standard pre-training converges to the invariant/minimizer set remains unproven.  The paper offers only empirical evidence, leaving the theoretical guarantee unresolved.

- **Narrow empirical scope:**  Verification is done in a single toy domain (5 state Boyans chain) and one architecture choice (30-layer looped linear Transformer).  No ablations on depth, context length, task scale are provided. The assertion that the experiments “empirically verify our theoretical insights” (in Introduction) is too strong given their limited breadth and the manually picked hyperparameters chosen to avoid numerical instability.

- **Occasional over-claiming of novelty:**  Some phrasing suggests an ever first link between standard RL losses and ICPE, whereas Wang et al. 2025 already established the connection for single-layer models; the manuscript would benefit highly from cooling down the claims

---

> ### Author Rebuttal · Authors · 2025-07-30
>
> We sincerely thank the reviewer for the comprehensive review and questions. We appreciate the reviewer's opinions on our theoretical rigour and the timeliness and importance of our work. We hope our response below addresses the reviewer's concerns and questions.
>
> > The key open question from Wang et al. whether standard pre-training converges to the invariant/minimizer set remains unproven.
>
> We agree that a convergence proof would significantly strengthen this work, and we have been considering this for a long time. It might not seem too difficult at first glance, given the success of the convergence proof of in-context regression in [2] and [3]. However, we want to highlight the unique challenge here -- the lack of the Polyak-Lojasiewicz (PL) inequality. Since the Transformer is highly nonlinear, PL is probably the only effective tool to use. Indeed, it's used in both [2] and [3]. Yet, we do not see any application of PL inequality in analyzing value-based RL algorithms. They are fundamentally different because regression is gradient descent (GD), whereas TD is not. TD is only semi-gradient. So, some nice symmetric property for establishing PL in regression does not hold in TD. Consequently, given the tremendous difficulty, the convergence proof, if it exists in the first place, deserves a project of its own. For example, [4] is the first paper to demonstrate the global optimality, and [2] is the first to show convergence. Both are very lengthy, even though they only studied regression. Our TD setting will inevitably make it more complicated.
>
> > No ablations on depth, context length, task scale are provided. The assertion that the experiments “empirically verify our theoretical insights” (in Introduction) is too strong ...
>
> We appreciate the reviewer for providing us with actionable suggestions. Before we delve into the additional results, we would like to note that during evaluation, we assessed the Transformers trained on 5-state MRPs across a range of context lengths (5-100). We believe the scalability of performance against the context length is a key and distinct feature of in-context learning. Thus, the reviewer's advice "...mismatch settings -- train with short context, evaluate with long" is precisely what we did for evaluating the Transformers' in-context policy evaluation capability.
> In light of the reviewer's suggestions, we did the following additional hyperparameter studies:
> - Depth: Keeping everything else unchanged, we additionally trained and evaluated 5-, 10-, and 60-layer Transformers for 20,000 tasks. We also repeated each experiment with 20 random seeds for statistical rigour. As we cannot upload figures during the rebuttal period, we can only verbally confirm that the pattern of $\theta^{TD}$ emerged in all the depths we tested, with the 60-layer Transformers having slightly more pronounced patterns. Let m denote the context length. In terms of quantitative evaluations, we paste here the approximate mean MSVEs on the shortest context (m = 5) and the longest context (m = 100) tested across 5-, 10-, and 60-layer Transformers trained via multi-task TD and MC.
>
>     Multi-task TD:
>
>     |   | 5-layer | 10-layer | 60-layer |
>     |---|---------|----------|----------|
>     | m = 5| 12.0 | 7.0 | 2.1 |
>     | m = 100 | 7.0 | 4.8 | 1.1|
>
>     Multi-task MC:
>     |   | 5-layer | 10-layer | 60-layer |
>     |---|---------|----------|----------|
>     | m = 5| 10.9| 7.8 | 3.0 |
>     | m = 100 | 6.3 | 5.3 | 1.9
>
>     We observe that in all cases, the mean MSVE dropped significantly as the context length increased, confirming the emergence of ICPE. Additionally, we witness that the deeper the Transformers, the better the prediction. Since under $\theta^{TD}$, one loop over the attention layer is equivalent to one step of TD update, we conjecture that the deeper Transformer applies more steps of TD updates internally, thus exhibiting better convergence. Lastly, we observe that the magnitudes of the mean MSVEs are of the same order between multi-task TD and MC. This observation provides further evidence that the Transformers trained via multi-task TD and MC are implementing the same in-context policy evaluation algorithm, at least behaviorally. It aligns with our theoretical result that $\theta^{TD}$ is a global minimizer of the NEU loss for both multi-task TD and MC.
>
> - Context length/State number/Feature dimension: Because of our construction, the number of states of the MRP, the context length, and the feature dimension are equal for training. For this study, we repeated our experiment on a 3-state Boyan's chain and a 10-state Boyan's chain for 20 random seeds. Limited by compute, we cannot make the MRPs too large due to the increased Transformer parameter size and Monte Carlo rollouts. We can confirm the emergence of $\theta^{TD}$ for both MRPs. Below is a table of the mean MSVEs from the quantitative evaluation.
>
>     Multi-task TD:
>
>     |   | 3-state MRP | 10-state MRP |
>     |---|---------|----------|
>     | m = 5| 2.9| 3.3 |
>     | m = 100 | 2.3 | 1.9 |
>
>
>     Multi-task MC:
>
>     |   | 3-state MRP | 10-state MRP |
>     |---|---------|----------|
>     | m = 5| 6.0| 3.8 |
>     | m = 100 | 4.75 | 1.9 |
>
>     The consistent decrease in MSVE confirms the in-context policy evaluation capability of the Transformer in both MRPs.
>
> - MRP: We further designed an MRP called Loop as follows: Given states $s_0, s_1, \dots, s_{n-1}$, $s_i$ can transition to $s_{i+1}$ for $i = 0, 1, \dots, n-2$. Furthermore, $s_{n-1}$ can transition to $s_0$, forming a closed loop. For each pair of $i, j$ where $j \notin \\{ i, ((i+1) \mod n)\\}$, we sample a number $k(i,j)$ between 0 and 1 uniformly. Given a threshold $\tau \in [0,1]$, $s_i$ can transition to $s_j$ if and only if $k(i, j) \ge \tau$. We then generate the transition probabilities for each state, retaining the connectivity. With this design, we guarantee that there exists only one communicating class due to the loop structure. Furthermore, we can choose $\tau$ to control the "hops" between states. We will also add the pseudocode for generating the Loop MRP in the next version of the manuscript. In this experiment, we chose $\tau = 0.5$ and ran the same training and validation procedures as in Boyan's chain experiments. We observed a clear $\theta^{TD}$ pattern for both multi-task TD and MC. For validation, we have the following table.
>
>     | |Multi-task TD | Multi-task MC |
>     |---|---------|----------|
>     | m = 5| 2.2| 3.5 |
>     | m = 100 | 0.3 | 1.1 |
>
>     Therefore, the numerical results suggest the emergence of ICPE capability of the Transformers when trained on this Loop MRP.
>
> We would also tone down our claim "empirically verifying our theoretical insights" to something more conservative, such as "preliminary empirical studies demonstrate overall consistency with our theoretical results".
>
> > Some phrasing suggests an ever first link between standard RL losses and ICPE, whereas Wang et al. 2025 already established the connection for single-layer models; the manuscript would benefit highly from cooling down the claims.
>
> We will update our claims to be more prudent to address the reviewer's concerns about overclaiming. We also thank and welcome the reviewer to follow up with other possible overclaims in our manuscript. In our defence, we wish to highlight that Wang et al., 2025 [1] justified the emergence of $\theta^{TD}$ by showing that it lies in an invariance set of the expected version of the multi-task TD update in the single-layer case only. Nevertheless, they did not connect $\theta^{TD}$ to any well-defined loss as in our analysis.
>
> [1] Jiuqi Wang, Ethan Blaser, Hadi Daneshmand, & Shangtong Zhang (2025). Transformers Can Learn Temporal Difference Methods for In-Context Reinforcement Learning. In The Thirteenth International Conference on Learning Representations.
>
> [2] Ruiqi Zhang, Spencer Frei, & Peter L. Bartlett (2024). Trained Transformers Learn Linear Models In-Context. Journal of Machine Learning Research, 25(49), 1–55.
>
> [3] Khashayar Gatmiry, Nikunj Saunshi, Sashank J. Reddi, Stefanie Jegelka, & Sanjiv Kumar (2024). Can Looped Transformers Learn to Implement Multi-step Gradient Descent for In-context Learning?. In Forty-first International Conference on Machine Learning.
>
> [4] Kwangjun Ahn, Xiang Cheng, Hadi Daneshmand, & Suvrit Sra (2023). Transformers learn to implement preconditioned gradient descent for in-context learning. In Thirty-seventh Conference on Neural Information Processing Systems.

---

> > ### Comment · Reviewer_De38 · 2025-08-03
> >
> > Thanks for the experiments and the discussion on the convergence proof, the latter is quite educational (love it!) and useful.
> >
> > I still finding the empirical results to be restricted to toy-settings, which is understandable given the limitations of the overall setup and paper scope. I will keep my score as is. However, I am positive about the paper and will support the acceptance in the case of a borderline situation.

---

> > > ### Author Response · Authors · 2025-08-05
> > >
> > > We thank the reviewer again for the constructive feedback that helped us improve our paper and the positive remarks about our work.

---

### Note · Authors · 2025-08-11

We use this opportunity to thank all the reviewers for their honest and constructive feedback.

We are pleased to learn the strengths of our work:
- We rigorously advance the theory of the emergence of in-context TD by proving that the ICTD parameters are a global minimizer of a well-defined loss (NEU), and tighten the connection between TD and MC targets in reinforcement pretraining.
- Our empirical study closely reproduces the predicted behavior.
- Our paper is well-structured and easy to follow.
- We clearly define the scope and limitations of our setup.

We also briefly summarize the revisions made to our paper in light of the suggestions:
- additional hyperparameter study on the Transformer depth, context length and the MRP
- Adjustment of claims of contributions
- resolving notation inconsistencies
- including explicit definition of multi-task TD and MC
- A more comprehensive survey of related works with an emphasis on applications
- repeated empirical evaluation with contexts built from unrolled trajectories rather than enumerated states
- a more explicit distinction between the settings of pretraining and evaluation to avoid confusion

We believe these revisions have significantly improved our paper. We are very grateful for the input from all the reviewers.

---

### Decision · Program_Chairs · 2025-09-17

**Decision:**

Accept (poster)

**Comment:**

The theoretical results are judged sound and interesting by all reviewers. However, the paper has also some weaknesses:
- Narrow empirical scope (Reviewer De38), limited practical applicability (Reviewer zvwM), "practical relevance (...) is still open" (Reviewer N5DC), "not representative of core challenges in modern RL" and "lacks scalability" (Reviewer xY5j).
- Some open questions are left for future work, for instance the "remaining convergence gap" (Reviewer De38).

The overall contribution is still appreciated despite the weaknesses. Given the consensus from the reviews with all scores being 4 (weak accept), the recommendation is to accept the paper.